# Subgenic Pol II interactomes identify region-specific transcription elongation regulators

Kevin M Harlen & L Stirling Churchman[*] 

## Abstract

Transcription, RNA processing, and chromatin-related factors all interact with RNA polymerase II (Pol II) to ensure proper timing and coordination of transcription and co-transcriptional processes. Many transcription elongation regulators must function simultaneously to coordinate these processes, yet few strategies exist to explore the complement of factors regulating specific stages of transcription. To this end, we developed a strategy to purify Pol II elongation complexes from subgenic regions of a single gene, namely the 5′ and 3′ regions, using sequences in the nascent RNA. Applying this strategy to *Saccharomyces cerevisiae,* we determined the specific set of factors that interact with Pol II at precise stages during transcription. We identify many known region-specific factors as well as determine unappreciated associations of regulatory factors during early and late stages of transcription. These data reveal a role for the transcription termination factor, Rai1, in regulating the early stages of transcription genome-wide and support the role of Bye1 as a negative regulator of early elongation. We also demonstrate a role for the ubiquitin ligase, Bre1, in regulating Pol II dynamics during the latter stages of transcription. These data and our approach to analyze subgenic transcription elongation complexes will shed new light on the myriad factors that regulate the different stages of transcription and coordinate co-transcriptional processes.

**Keywords** interactomes; MS2 stem-loops; proteomics; RNA polymerase II; transcription

**Subject Categories** Chromatin, Epigenetics, Genomics & Functional Genomics; Methods & Resources; Transcription

**Mol Syst Biol. (2017) 13: 900**

## Introduction

Production of proper mRNA transcripts occurs through three highly regulated stages of transcription. Transcription is initiated after recruitment of RNA polymerase II (Pol II) to the promoter. After promoter escape, Pol II enters the elongation stage as it transcribes through the gene body. Finally, transcription is completed during termination when the transcript is processed and Pol II is released from the chromatin template (Perales & Bentley, 2009; Hahn & Young, 2011; Kuehner *et al*, 2011; Rando & Winston, 2012; Kwak & Lis, 2013). Transcription elongation itself is a highly regulated process, consisting of multiple stages (Jonkers & Lis, 2015). In metazoans, promoter proximal pausing separates early and productive elongation in a tightly regulated manner (Adelman & Lis, 2012). During promoter proximal pausing, many factors are recruited to Pol II in an effort to transition the elongation complex (EC) from early into productive elongation. While yeast does not have a sharp promoter proximal pause, transition of yeast ECs into productive elongation is also regulated (Pokholok *et al*, 2002; Mayer *et al*, 2010; Lidschreiber *et al*, 2013; Rodríguez-Molina *et al*, 2016).

Transcription elongation is also controlled to ensure proper timing of co-transcriptional processes which are most pronounced at the early and late stages of transcription elongation (Perales & Bentley, 2009; Hsin & Manley, 2012; Bentley, 2014; Fusby *et al*, 2015). In the early stages of elongation, Pol II must recruit factors to transition into productive elongation and perform the necessary capping of mRNA, which is required for the downstream RNA processing events of splicing and 3′ end processing (Cooke & Alwine, 1996; Schwer & Shuman, 1996; Flaherty *et al*, 1997; Topisirovic *et al*, 2011). In the latter stages of elongation, Pol II must recruit 3′ end processing factors as well as factors that regulate the transition from elongation to termination.

Dynamic phosphorylation of the Pol II C-terminal domain (CTD) facilitates the arrival and dissociation of many key regulatory factors at the right time and place (Buratowski, 2009; Hsin & Manley, 2012; Jeronimo *et al*, 2013; Bentley, 2014). The CTD consists of a conserved heptapeptide repeat $Y_1S_2P_3T_4S_5P_6S_7$ that is repeated 26 times in yeast, and phosphorylation of Tyr1, Ser2, Thr4, Ser5, and Ser7 has distinct patterns of enrichment across gene bodies. For example, Ser5P is highly enriched early in transcription and as such interacts with many factors involved in early elongation (Komarnitsky *et al*, 2000; Schroeder *et al*, 2000; Schwer & Shuman, 2011; Harlen *et al*, 2016). However, phospho-CTD patterns are not entirely unique to one region of the gene. For instance, Ser5P also peaks near 3′ splice sites (Harlen *et al*, 2016) and Ser5P and Ser2P are broadly associated with Pol II (Schüller *et al*, 2016; Suh *et al*, 2016). Thus, while the CTD phosphorylation state serves as a proxy for transcription stage, it does not dictate precise boundaries during transcription.

Department of Genetics, Harvard Medical School, Boston, MA, USA
*Corresponding author. Tel: +1 617 432 1917; E-mail: churchman@genetics.med.harvard.edu

Understanding which factors regulate these processes is key to determining how Pol II regulates myriad co-transcriptional processes at precise regions across the gene body. Chromatin immunoprecipitation (ChIP) has been a powerful way to localize individual factors along gene bodies, which likely reports on where the factor functions. However, ChIP can only report on a single factor at one time, thus unraveling regulatory mechanisms that use multiple factors together can be laborious. Moreover, ChIP-based approaches are biased, as they require a prior appreciation for a factor's role in transcription regulation.

We sought to develop an unbiased and comprehensive strategy to identify the factors that associate with Pol II at specific regions of a gene. We chose to isolate Pol II elongation complexes (ECs) based on the emerging nascent RNA, allowing the purification of complexes from precise genomic locations. We tested this strategy by developing a system to isolate ECs from the two most distinct stages of elongation: (i) early productive elongation and (ii) late-stage elongation just prior to termination. We used two RNA stem-loops, PP7 and MS2, which bind tightly to viral coat proteins (Bardwell & Wickens, 1990). These sequences have been used to study Pol II elongation and splicing dynamics (Lacadie *et al*, 2006; Larson *et al*, 2011; Hocine *et al*, 2013; Lenstra *et al*, 2015). By inserting just two repeats of the PP7 or MS2 stem-loop sequences into the UTRs of a single gene and co-expressing the coat binding proteins, we are able to isolate ECs from early and late stages of transcription. Analysis of EC composition by quantitative mass spectrometry yields the respective Pol II interactomes from early and late elongation. Interestingly, both the early and late ECs contain factors that were not previously appreciated to function during those stages. Through the high-resolution mapping of Pol II by native elongating transcript sequencing (NET-seq) (Churchman & Weissman, 2011), we investigated the region-specific effects of these factors. Indeed, we find a role for transcription termination factor, Rai1, in regulating early transcription elongation genome-wide and a novel role for the histone modifier, Bre1, in regulating Pol II dynamics at polyA sites. These results demonstrate how isolation of distinct Pol II complexes can be used to determine the transcriptional regulatory factors governing different stages of transcription elongation.

## Results

### Isolation of region-specific transcription complexes

To explore which factors regulate the early and late stages of transcription, we developed a strategy to isolate region-specific transcription elongation complexes (ECs). We designed a synthetic construct that allowed isolation of ECs from the 5′ or 3′ end of a single gene (Fig 1A). The *Saccharomyces cerevisiae TDH3* gene flanked upstream by two PP7 RNA stem-loop encoding sequences and downstream by two MS2 RNA stem-loop encoding sequences was placed under control of an inducible *GAL* promoter. This construct was cloned into a high-copy plasmid and transformed into yeast along with a second plasmid constitutively expressing either a PP7 coat binding protein (PCP) fused to GFP or a MS2 coat binding protein (MCP) fused to mCherry (RFP). After inducing expression by addition of galactose to the media, isolation of ECs from either the 5′ or 3′ end of this construct occurs via a sequential purification. First, all Pol II complexes are purified using an epitope tag on the Rpb3 subunit of Pol II (Fig 1A and B, Rpb3 input/unbound). While still bound to beads, the nascent RNA is fragmented by RNase A digestion followed by specific elution of the complexes. Next, a second highly efficient IP (Fig 1A and B, GFP or RFP input/unbound) is performed targeting the coat proteins bound to the hairpins: anti-GFP antisera targeting the PCP-GFP bound to the PP7 stem-loops at the 5′ end of the transcript or anti-RFP antisera targeting the MCP-RFP bound to the MS2 stem-loops at the 3′ end of the transcript. The doubly purified complexes are then analyzed by tandem mass spectrometry to identify factors associated with either 5′ or 3′ ECs (Fig 1A).

We tested the ability of the sequential IP to isolate ECs by comparing a single Rpb3 IP, the sequential Rpb3-GFP IP, and a single GFP IP that is expected to isolate mainly mature RNP complexes. When analyzed by silver stain, the proteins purified by the sequential Rpb3-GFP IP have a banding pattern that is highly similar to the Rpb3 IP but largely different from the single GFP IP banding pattern (Fig 1C). These data demonstrate the ability of the sequential IP to isolate actively transcribing Pol II ECs.

Next, we optimized our approach to ensure isolation of complexes enriched at either the 5′ or 3′ end of the gene. The RFP IP targeting the 3′ end of the transcript can only purify complexes located at the 3′ end of the gene after transcription of the MS2 stem-loops. However, the GFP IP can target complexes at any point after transcription of the PP7 stem-loops placed in the 5′ UTR. To restrict our analysis to the 5′ end of the gene, complexes purified by the Rpb3 IP were treated with RNase A while still bound to beads in order to fragment the nascent RNA, producing ECs with short pieces of nascent RNA. Thus, complexes proximal to the 5′ UTR will be labeled by the PP7 stem-loops in the nascent RNA and complexes further downstream will not. Different intervals of RNase treatment were tested to find the optimal fragmentation time to enrich 5′ transcript regions over 3′ transcript regions after the GFP IP (Fig 1D). Analysis of sequential Rpb3-GFP and Rpb3-RFP IPs by silver stain displayed similar banding patterns to one another and to the Rpb3 IP alone; however, differences were detectable (Fig 1C and E). These data suggest sequential IPs targeting the 5′ or 3′ ends of genes purify Pol II complexes. Lastly, to ensure a quantitative purification of ECs, all IPs were optimized to be highly efficient (Fig 1B).

As a final test of the ability of our sequential IP system to isolate ECs from a specific gene and from subgenic regions, we performed NET-seq on complexes from the sequential Rpb3-GFP and Rpb3-RFP isolations. NET-seq maps actively transcribing Pol II at nucleotide resolution (Churchman & Weissman, 2011), allowing the precise localization of the ECs isolated. First, NET-seq analysis revealed that the targeted transcript was highly enriched over non-targeted Pol II transcripts (Fig 1F). Second, cumulative analysis of NET-seq reads following a sequential GFP (5′) IP revealed that > 60% of isolated complexes were located in the first 400 base pairs (bp) of the gene. In contrast, greater than half of complexes isolated by sequential RFP (3′) IPs were located in the last 200 bp of the gene (Fig 1G). These data demonstrate the ability of our approach to enrich for ECs from different regions of a single gene.

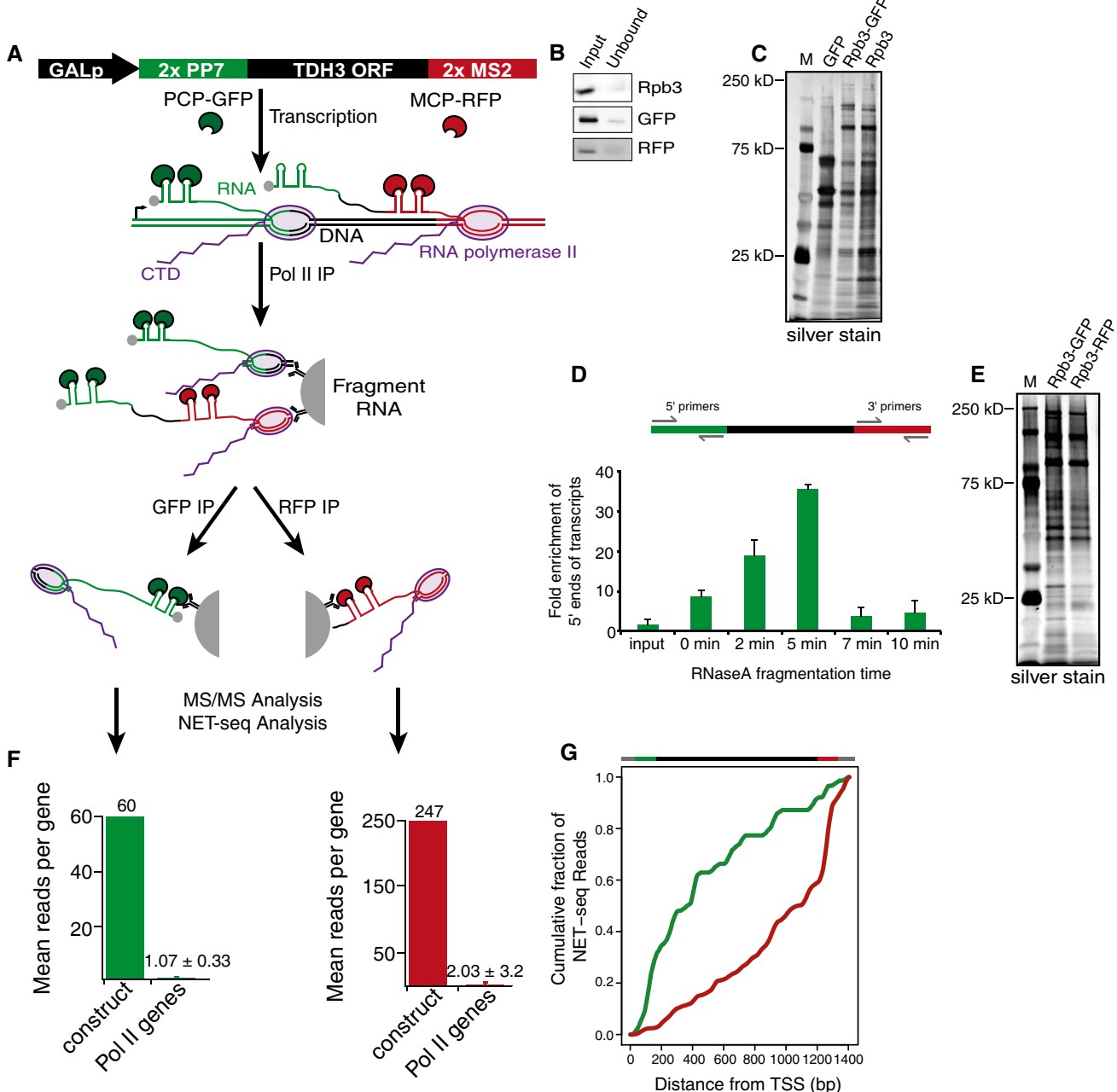

**Figure 1.  Isolation of transcription elongation complexes from the 5′ and 3′ regions of a single gene.**

A   Schematic demonstrating the sequential immunoprecipitation (IP) approach used to isolate 5′ and 3′ elongation complexes (ECs). GALp, GAL1 promoter; 2× PP7, two repeats of the PP7 stem-loop encoding sequence; 2× MS2, two repeats of the MS2 stem-loop encoding sequence; PCP-GFP, PP7 coat protein fused to GFP; MCP-RFP, MS2 coat protein fused to RFP.

B   Western blots for input and unbound samples for Rpb3 (Pol II), GFP (5′ complexes), and RFP (3′ complexes) immunoprecipitations. The input samples for the GFP and RFP IPs correspond to the bound fraction of the Rpb3 IP.

C   Silver stain of ECs isolated by single GFP IP (GFP), Pol II-GFP sequential IP (Rpb3-GFP), and single Pol II (Rpb3) IPs. M, molecular weight marker; kD, kilodalton.

D   Top: depiction of RT–qPCR primer location for 5′ region and 3′ region primers in the stem-loop-*TDH3* construct. Bottom: RT–qPCR demonstrating the fold enrichment of 5′ ECs over 3′ ECs after various RNase A fragmentation times. The input sample represents whole-cell lysate while the 0 min sample is taken after the Rpb3 IP with no RNase A fragmentation.

E   Silver stain of 5′ (Rpb3-GFP) and 3′ (Rpb3-RFP) ECs. M, molecular weight marker; kD, kilodalton.

F   Bar graphs displaying the average number of reads per gene from NET-seq analysis on sequentially isolated 5′ and 3′ ECs. Construct indicates reads from the stem-loop-*TDH3* construct; Pol II genes indicates reads from all other Pol II genes. Error bars for Pol II genes represent standard deviation. 186 Pol II genes were detected in the 5′ (GFP) IPs, and 1,118 Pol II genes were detected in the 3′ (RFP) IPs.

G   Top: depiction of the stem-loop-*TDH3* construct. Bottom: cumulative fraction of NET-seq reads plotted as a function distance from the transcription start site of the stem-loop-*TDH3* construct for 5′ (green) and 3′ (red) ECs.

## Analysis of 5′ and 3′ transcription elongation complexes

To identify the composition of early and late ECs, 5′ targeted, 3′ targeted, and mock IPs were carried out in biological triplicate and were analyzed by tandem mass spectrometry to identify proteins present in each sample (Table EV1). Importantly, the mock IP is conducted after the first Pol II IP, representing non-specific interactions of all Pol II interactors (Fig 2A–C). Analysis of normalized MS1 intensities (see Materials and Methods and Fig EV1 for details) between all samples shows triplicate IPs of 5′, 3′, and mock ECs are highly reproducible with an average Pearson correlation of 0.9, while comparisons of 5′ to mock, 3′ to mock, and 5′ to 3′ are less consistent with an average Pearson correlation of 0.77. Principal component analysis confirms the reproducible and distinct composition of each EC because three unique clusters of biological triplicate samples are present for 5′, 3′, and mock ECs (Fig 2B). Lastly, we performed a complete linkage hierarchical clustering analysis of purified factors that grouped 5′ and 3′ ECs separately from the mock IP and split samples into 5′ and 3′ ECs (Fig 2C), demonstrating that both 5′ and 3′ ECs contain interactomes unique from the mock IP and from one another.

Factors enriched in either 5′ or 3′ ECs samples were determined using quantitative label-free mass spectrometry analysis (Tables EV1 and EV2) (Hubner *et al*, 2010; Hubner & Mann, 2011). Volcano plots were determined by calculating the average fold change in

normalized MS1 spectral intensities of each protein plotted against the *P*-value derived from *t*-tests between the two samples (Fig 3A–C). Only proteins present in all three biological samples were used for analysis. This label-free analysis allows a direct comparison between the relative abundance of each protein in the 5′, 3′, or mock datasets. A false discovery rate (FDR) of either 0.05, as a stringent cutoff for high-confidence interactors, or 0.1, as less stringent cutoff, was used to determine significantly enriched interactors (black, green, or red diamonds in Fig 3A–C). An emphasis on a low FDR inevitably results in false negatives, so when a factor is not significantly enriched that does not imply it does not interact specifically with Pol II in the particular subgenic region.

Analysis of the significant interactors within the 5′ region of the gene (Fig 3A, and Tables EV1 and EV2) revealed many known transcription elongation factors, including the mRNA capping complex, the Paf1 complex, the Spt4/5 complex and Spt6, as among the most enriched proteins (Table EV2). Many of these factors have been implicated in regulating the early stages of productive elongation in both yeast and mammals (Pokholok *et al*, 2002; Mayer *et al*, 2010; Lidschreiber *et al*, 2013; Chen *et al*, 2015; Yu *et al*, 2015). Factors enriched in 3′ complexes included the CCR4-NOT complex (Kruk *et al*, 2011; Dutta *et al*, 2015) and the CTD prolyl isomerase, Ess1, which has been shown to be a positive regulator of transcription termination (Fig 3B) (Wu *et al*, 2003; Krishnamurthy *et al*, 2009; Ma *et al*, 2012). Our stringent statistical analysis likely results in a

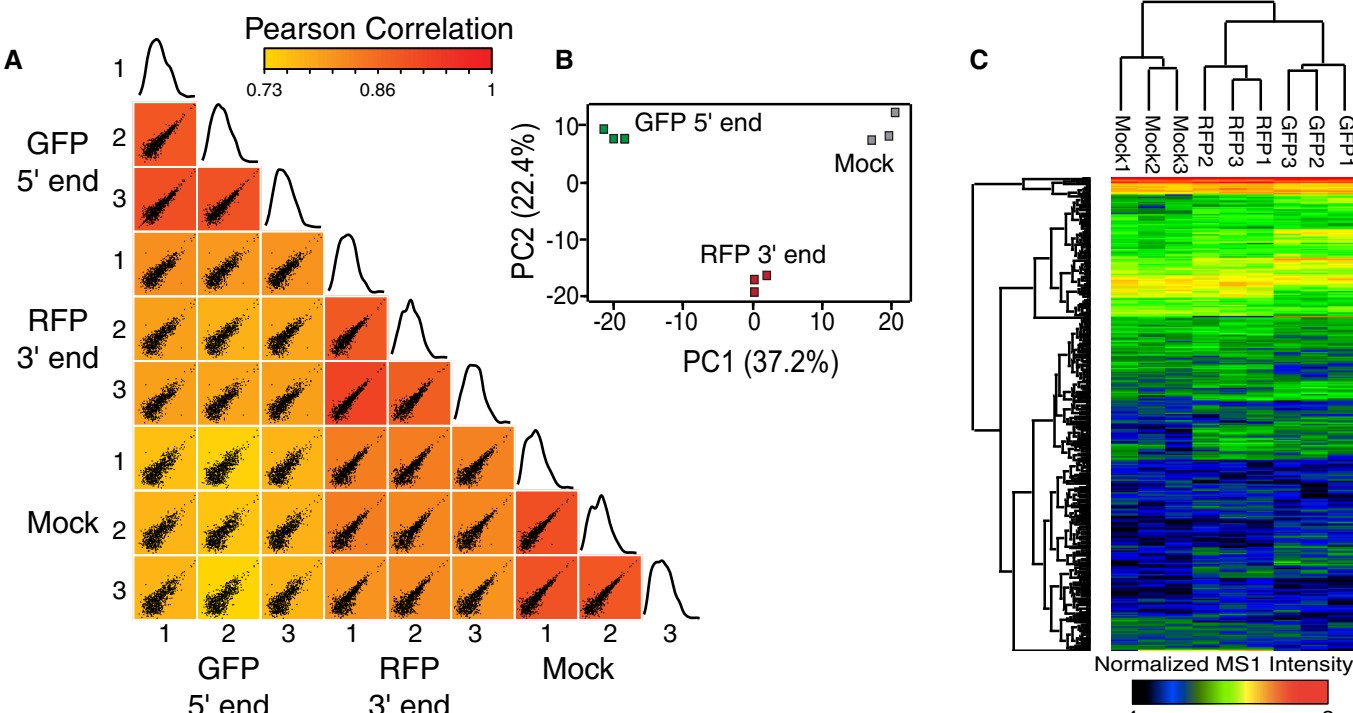

**Figure 2.  5′ and 3′ elongation complexes are distinct.**

A   Scatter plots comparing the normalized log₂ MS1 intensities for each protein from triplicate GFP (5′), RFP (3′), and mock IPs. Plots are colored based on the Pearson correlation value between the samples. The last plot in each row depicts a histogram of MS1 intensities for all proteins in each sample. MS1 intensities for each peptide from a given protein were summed to obtain MS1 intensity levels for each protein.

B   Principal component analysis of triplicate GFP (5′), RFP (3′), and mock IPs. PC, principal component.

C   Complete linkage hierarchical clustering of normalized MS1 intensities for triplicate GFP (5′), RFP (3′), and mock samples.

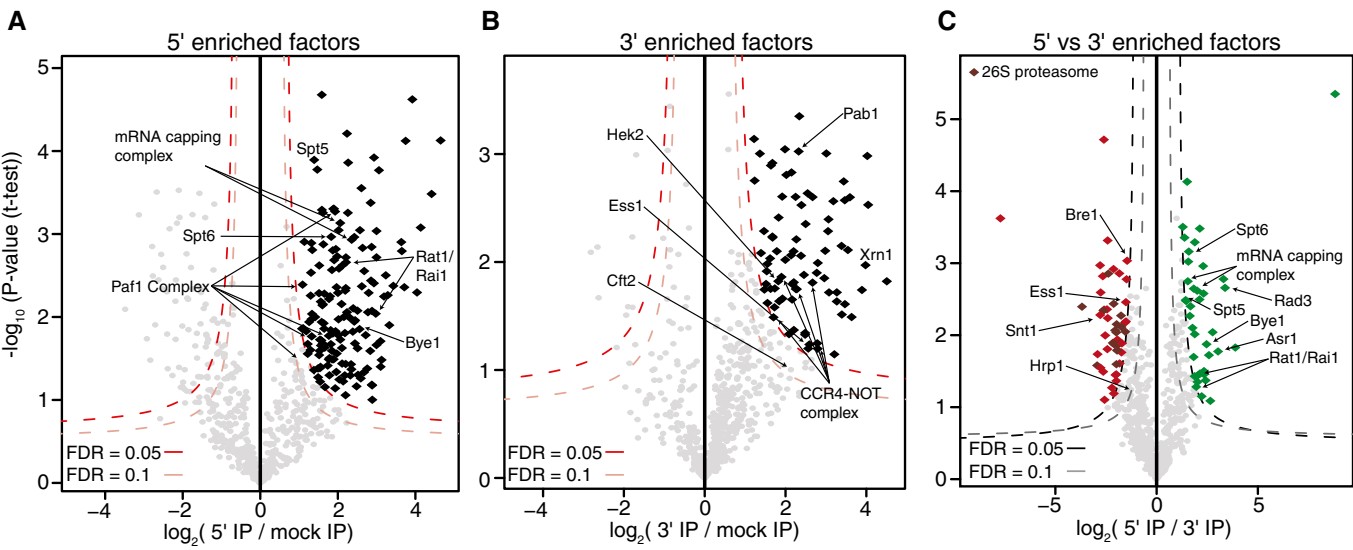

**Figure 3.  Identifying factors enriched in 5′ and 3′ elongation complexes.**

A, B  Volcano plot analysis comparing GFP (5′) IPs (A) or RFP (3′) IPs (B) to mock IPs. High-confidence, specifically enriched factors determined using an FDR of 0.05 and less stringent interactors determined using an FDR of 0.1. Factors mentioned in the text are labeled.

C  Volcano plot comparing GFP (5′) and RFP (3′) IPs. Factors enriched in the 5′ complexes are colored in green, and those enriched in the 3′ complexes are colored in red. Components of the 26S proteasome are colored in maroon; factors mentioned in the text are labeled.

number of false negatives. For example, some components of the cleavage and polyadenylation machinery, such as Pcf11, were enriched in the 3′ IP, but were not determined statistically significant. Additionally, 3′ cleavage factors likely experience a transient association with Pol II before interacting with the nascent RNA. Nevertheless, Cft2 and Pab1, components of the cleavage and polyadenylation machinery, were also enriched in the 3′ IP, demonstrating that our approach is capable of identifying 3′ RNA processing factors.

Lastly, comparison of the 5′ and 3′ enriched factors again confirmed the region-specific IPs (Fig 3C). The mRNA capping complex, the DNA helicase of the general transcription factor, TFIIH, and transcription elongation factors, Spt5 and Spt6, were all enriched in the 5′ region of the gene compared to the 3′ region. Interestingly, 15 components of the 26S proteasome were enriched in the 3′ region. The proteasome has been implicated in transcriptional regulation, particularly at inducible genes (Gillette *et al*, 2004; Lee *et al*, 2005; Geng *et al*, 2012), consistent with the construct being under control of the *GAL1* promoter and induced by galactose. Together, these data demonstrate the unique capability of this system to isolate subgenic ECs.

**Global analysis of 5′ enriched factors, Bye1 and Rai1, confirms their role in early transcription elongation**

Enriched in the 5′ but not 3′ complexes, Bye1 is a negative regulator of transcription elongation that binds to both Pol II and trimethylated lysine 4 on histone H3 (H3K4me3) (Venters *et al*, 2011; Kinkelin *et al*, 2013). ChIP profiles of Bye1 also confirm its enrichment near the 5′ region of genes with no signal near the 3′ end of genes (Kinkelin *et al*, 2013; Pinskaya *et al*, 2014). NET-seq analysis of *bye1Δ* cells revealed that Pol II density at the 5′ region of the

*RPL26B* gene as well as throughout the gene body was severely affected (Fig 4A), but no defects were seen at the 3′ end of the gene near the polyA site. The same pattern was observed genome-wide where loss of *BYE1* causes defects in both early and productive elongation but does not affect Pol II dynamics near polyA sites (Fig 4B). When compared to WT cells, the defect in Pol II dynamics in *bye1Δ* cells significantly reduced the density of Pol II near the promoter (Fig 4C) and increased Pol II density throughout gene bodies. This is consistent with Bye1 serving as negative regulator of transcription elongation in the early stages of transcription (Kinkelin *et al*, 2013; Pinskaya *et al*, 2014). These data demonstrate the ability of region-specific IPs to enrich and identify factors affecting Pol II transcription at specific regions along gene bodies.

We next used the 5′ and 3′ elongation complex data to explore new, region-specific roles of unappreciated or novel transcription elongation factors. Surprisingly, analysis of the 5′ enriched complexes displayed enrichment for the Rat1/Rai1 exonuclease complex. While the major function of this complex is in transcription termination (Kim *et al*, 2004), the essential exonuclease, Rat1, has been suggested to function as an elongation factor at some genes in yeast (Jimeno-González *et al*, 2010) and its mammalian homologue, Xrn2, has been implicated in regulating early transcription dynamics near promoter proximal pause sites (Brannan *et al*, 2012). To test the possibility that Rat1/Rai1 functions in the early stages of transcription, we re-analyzed NET-seq data from *rai1Δ* cells (Harlen *et al*, 2016), focusing on the 5′ region of genes. As a control, we also re-analyzed cells lacking a third termination factor, Rtt103, that functions with Rat1/Rai1 in transcription termination (Kim *et al*, 2004; Harlen *et al*, 2016). Rtt103 was not enriched in the 5′ complexes. As determined in our previous work, loss of both Rai1 and Rtt103 caused defects in transcription termination, with loss of Rai1 inducing high levels of transcriptional read-through (Fig 4D

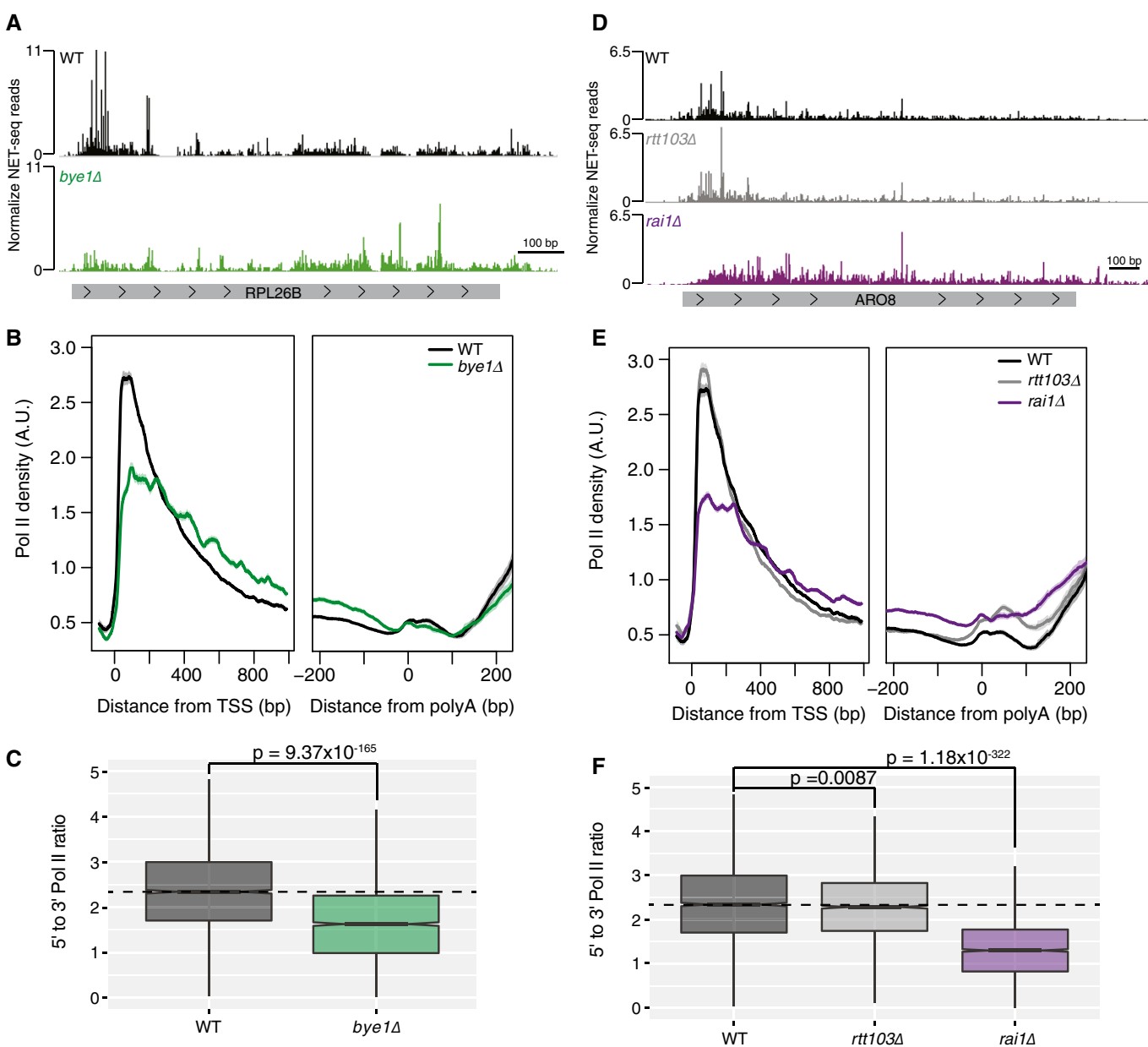

**Figure 4.  Bye1 and Rai1 function during early transcription elongation.**

A   Normalized NET-seq reads for WT (black) and *bye1Δ* (green) cells at the *RPL26B* gene.
B   Normalized average NET-seq profiles for WT and *bye1Δ* cells around the transcription start site (TSS) and polyadenylation site (polyA) of protein-coding genes.
C   Box plot comparing the ratio of Pol II density at the 5′ region of genes to the Pol II density at the 3′ region of genes for WT and *bye1Δ* cells at protein-coding genes.
D   Normalized NET-seq reads for WT (black), *rtt103Δ* (gray), and *rai1Δ* (purple) cells at the *ARO8* gene.
E   Normalized average NET-seq profiles for WT and *rtt103Δ*, and *rai1Δ* cells around the TSS and polyA site of protein-coding genes.
F   Box plot comparing the ratio of Pol II density at the 5′ region of genes to the Pol II density at the 3′ region of genes for WT, *rtt103Δ*, and *rai1Δ* cells at protein-coding genes.

Data information: In (B) and (E), NET-seq reads for each gene are normalized by total reads for each gene in the analyzed region, and shaded areas represent the 95% confidence interval, *n* = 2,738 genes. A.U., arbitrary units. In (C) and (F), *P*-values were determined using a two-sided *t*-test, *n* = 2,734 genes. WT, *rtt103Δ*, and *rai1Δ* NET-seq data were obtained from Harlen *et al* (2016) and re-analyzed in (D–F). In (C) and (F) horizontal bars indicate samples being compared by *t*-test. Dashed lines mark the median 5′ to 3′ Pol II ratio for WT cells. Solid lines in box plots represent sample median. Filled regions in box plots range form the 25th to 75th percentiles of the data while vertical lines are 1.5 times the inter-quartile range.

and E) (Harlen *et al*, 2016). However, while *rtt103Δ* did not affect Pol II dynamics in the early stages of transcription, *rai1Δ* resulted in dramatic changes to Pol II dynamics in the 5′ ends of genes (Fig 4D and E). Loss of Rai1 resulted in a loss of Pol II accumulation near

the promoter of the *ARO8* gene (Fig 4D), while loss of Rtt103 had no effect. Pol II density was also affected throughout gene bodies where Pol II complexes appeared to accumulate. These effects were observed genome-wide, as seen by analyzing average Pol II

occupancy in WT, rtt103Δ, and rai1Δ cells near the 5′ and 3′ regions of coding genes (Fig 4E). Comparison of Pol II density near the promoter with density near the polyA site revealed that rai1Δ, but not rtt103Δ cells, have a decreased 5′ to 3′ Pol II density ratio (Fig 4F), again consistent with RAI1 deletion resulting in defects in Pol II elongation near promoters. Interestingly, loss of Rai1 displayed similar effects to loss of Bye1, suggesting that in addition to its role in transcription termination Rai1 may also act as a negative regulator of transcription in the early stages of transcription elongation. To test this possibility, we assessed sensitivity of rai1Δ and bye1Δ to mycophenolic acid (MPA), a transcription elongation inhibitor. Both rai1Δ and bye1Δ displayed reduced sensitivity to MPA; in contrast to loss of the known transcription elongation factor DST1 (TFIIS) that displayed increased sensitivity to MPA (Fig EV2), consistent with both Bye1 and Rai1 functioning as negative transcriptional regulators. Together, our data indicate a role for Rai1, and perhaps the full Rat1/Rai1 exonuclease complex, as a genome-wide transcription elongation factor in yeast, with functions in both transcription elongation and termination.

### Bre1 functions in the 3′ region of genes during transcription

Analysis of factors enriched at the 3′ end of genes revealed many factors with known functions in transcription elongation, 3′ end processing, and termination (Fig 3B and C). Interestingly, the ubiquitin ligase, Bre1, was also enriched in 3′ ECs. Bre1, in conjunction with the ubiquitin-conjugating enzyme Rad6, monoubiquitinates both histone H2B on lysine 123 (H2Bub) and multiple residues on the cleavage and polyadenylation factor Swd2 (Crisucci & Arndt, 2011; Vitaliano-Prunier et al, 2012). Bre1 has been shown to be recruited to Pol II though the Paf1 complex (Crisucci & Arndt, 2011; Piro et al, 2012). Multiple Paf1 complex proteins were enriched in 5′ ECs, but none were enriched in the 3′ ECs. However, a lack of enrichment is surprising as subunits of the Paf1 complex have been shown to localize across gene bodies by ChIP (Crisucci & Arndt, 2011). Our stringent statistical analysis reduces false positives at a cost of false negatives. Indeed, Paf1 complex members, Paf1 and Ctr9, show a modest enrichment of 1.5-fold in the 3′ interactome with low P-values (0.018 and 0.013). To look for transcriptional defects brought on by loss of Bre1, we analyzed bre1Δ cells by NET-seq. Consistent with analysis of 5′ and 3′ ECs, loss of BRE1 caused minimal changes to Pol II density near the 5′ region of genes but did alter Pol II dynamics in the latter stages of transcription elongation and around polyA sites (Fig 5A and B). A lack of major changes to Pol II dynamics at the 5′ end of genes is somewhat surprising as H2Bub is important for proper deposition of other histone modifications such as H3K4 methylation (Batta et al, 2011; Crisucci & Arndt, 2011). However, at actively transcribed genes H2Bub is important for modulating transcription elongation (Xiao et al, 2005; Batta et al, 2011), consistent with the defects in transcription elongation observed by NET-seq in the bre1Δ mutant. Loss of BRE1 resulted in a significant increase in Pol II downstream of the polyA site at 60% of genes (Fig 5C); however, it did not lead to transcriptional read-through. These data suggest that Bre1 regulates Pol II pausing during the latter stages of transcription elongation and 3′ end processing. As Paf1 complex components, Paf1 and Rtf1, have been implicated in the recruitment of Bre1 to Pol II (Ng et al, 2003; Xiao et al, 2005; Piro et al, 2012), we analyzed both paf1Δ

and rtf1Δ cells by NET-seq. Rtf1 is a unique member of the Paf1 complex as Rtf1 alone can promote H2Bub through Rad6/Bre1 (Piro et al, 2012). Both paf1Δ and rtf1Δ mutants did induce minor alterations to Pol II density at polyA sites, consistent with the role of Paf1 complex in 3′ end processing (Mueller et al, 2004; Penheiter et al, 2005; Crisucci & Arndt, 2011). However, neither loss of PAF1 nor RTF1 resulted in increased Pol II pausing during polyadenylation as was seen with loss of BRE1 (Fig 5C), suggesting that Bre1 may regulate Pol II independent of the Paf1 complex at the 3′ end of genes.

Considering our findings, we postulated that H2Bub and ubiquitinated Swd2 might work together to regulate Pol II dynamics at the 3′ end of genes. Ubiquitination of Swd2 by Bre1 is important for proper 3′ end processing and export of mRNP complexes (Vitaliano-Prunier et al, 2012) and H2Bub is important for the reassembly of nucleosomes in the wake of transcription and promotes transcription elongation at actively transcribed genes (Batta et al, 2011). To ask whether ubiquitination of H2B occurs downstream of the polyA site, we re-analyzed published MNase-seq and histone modification ChIP-seq data (van Bakel et al, 2013; Rhee et al, 2014) that reveals the presence of a nucleosome just downstream of the polyA site that is enriched for H2Bub (Fig 5D). To confirm that the enrichment of H2Bub at the post-polyA nucleosomes is not due to contamination from a proximal +1 nucleosome of a downstream gene, we looked a subset of genes that are at least 350 bp away from another gene and observe a similar trend (Fig EV3). Together, these data suggest a role for Bre1 in regulating Pol II during the latter stages of transcription. This regulation may be mediated through its ubiquitin ligase activity directed toward H2B or through altering ubiquitination or regulation of other factors, such as Swd2, important for 3′ end Pol II dynamics. This analysis of transcriptional defects brought on by the bre1Δ mutations reveals how the 3′ EC data can be used to interrogate factors regulating transcription at the 3′ end of genes.

## Discussion

The coupling of transcription to co-transcriptional processes, such as RNA processing and chromatin modifications, requires Pol II to interact dynamically with a multitude of factors. Not only must the interactions occur during transcription, but they must occur at the proper stage of transcription. Thus, Pol II has to recruit a highly unique set of factors to modulate transcription initiation, elongation, and termination. While techniques like ChIP can be used to study a single factor across gene bodies, it cannot provide information on the suite of proteins regulating Pol II at a specific stage of transcription. To gain insight into how the complete composition of the Pol II elongation complex regulates the stages of transcription, we developed a technique to isolate ECs specifically from the 5′ (early elongation) and 3′ (late elongation/termination) regions of a gene. Purification of all Pol II complexes is followed by isolation of ECs at specific regions of a gene using sequences encoded in the nascent RNA. This allows precise purification of ECs from distinct 5′ and 3′ genic regions. Analysis of the isolated complexes by quantitative mass spectrometry determines the proteins functioning at the 5′ and 3′ regions of the gene, demonstrating the dynamic nature of the transcription elongation complex.

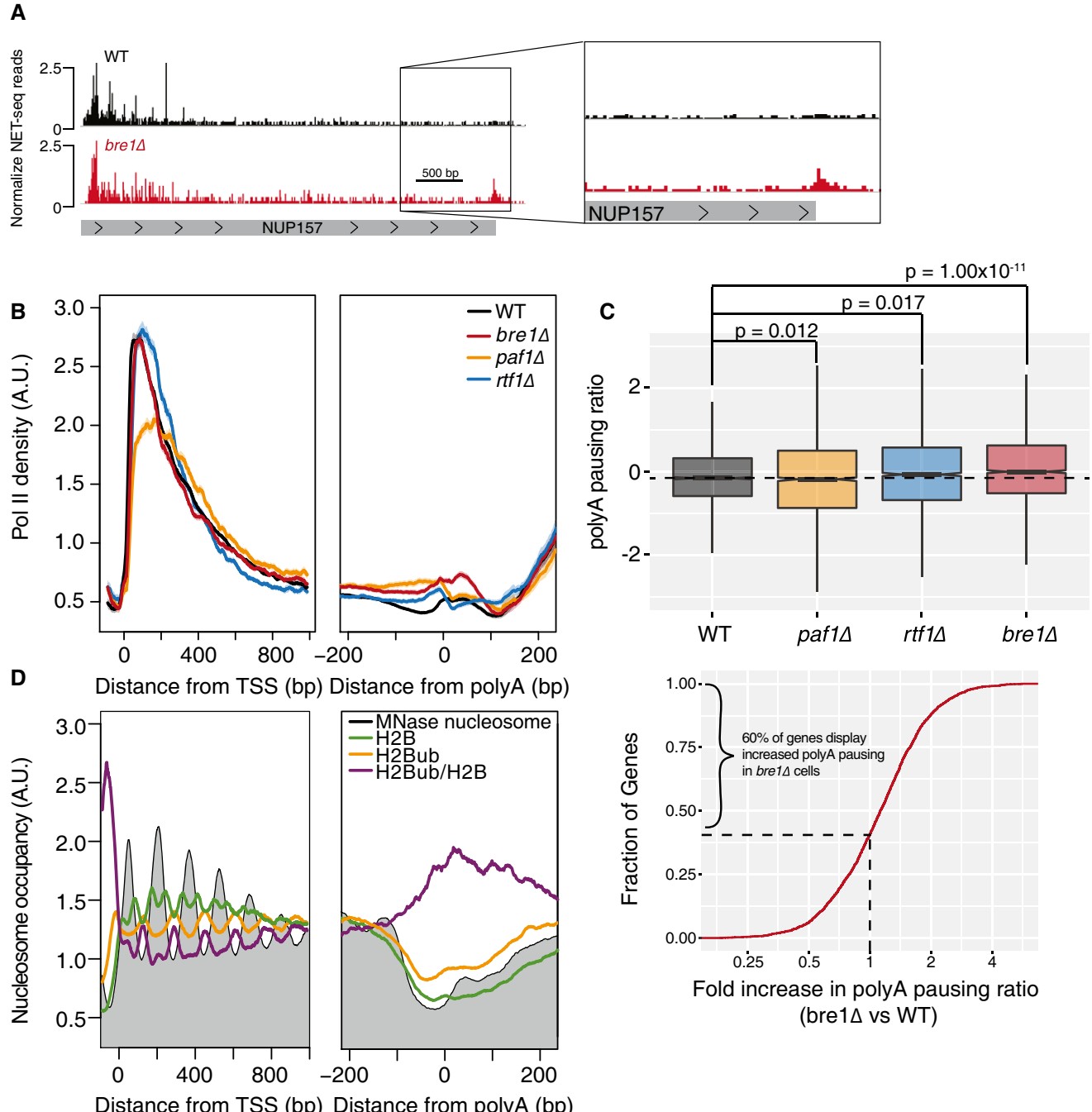

**Figure 5. Bre1 functions during the latter stages of transcription elongation.**

A   Normalized NET-seq reads for WT (black) and *bre1Δ* (red) cells at the *NUP157* gene. The panel on the right emphasizes the window around the polyA site where *bre1Δ* cells induce Pol II pausing.

B   Normalized average NET-seq profiles for WT and *bre1Δ*, *paf1Δ*, and *rtf1Δ* cells around the transcription start site TSS and polyA of protein-coding genes. NET-seq reads for each gene are normalized by total reads for each gene in the analyzed region; shaded areas represent the 95% confidence interval, n = 2,738 genes. A.U., arbitrary units.

C   Top: box plot comparing the $\log_2$ ratio of NET-seq reads in a window from the polyA site to 100 base pairs downstream of the polyA site to reads in a window of 50 base pairs upstream of the polyA site to the polyA site. *P*-values determined using a two-sided *t*-test, n = 2,674 genes. Horizontal bars indicate samples being compared by *t*-test. The dashed line marks the median polyA pausing ratio for WT cells. Solid lines in box plots represent sample median. Filled regions in box plots range form the 25th to 75th percentiles of the data while vertical lines are 1.5 times the inter-quartile range. Bottom: cumulative distribution of the fold change in polyA pausing ratio between *bre1Δ* and WT cells; 60% percent of genes display increased polyA pausing in *bre1Δ* cells compared to WT cells.

D   Average normalized MNase seq (gray) or ChIP-exo for H2B (green), H2B ubiquitylation (H2Bub, gold), and H2Bub/H2B (purple) around the TSS and polyA sites of protein-coding genes. MNase-seq and ChIP-exo reads are normalized by total reads for each gene in the analyzed region, n = 2,738. MNase-seq data were obtained from van Bakel *et al* (2013), and ChIP-exo data were obtained from Rhee *et al* (2014).

          

Our analysis of the 5′ Pol II EC interactome revealed many known early elongation factors and the surprising enrichment of termination factors, Rat1 and Rai1 (Fig 6). Rat1 has impacts on transcription elongation, and its mammalian homologue, Xrn2, has been implicated in promoter proximal pause regulation in mammalian cells (Jimeno-González *et al*, 2010; Brannan *et al*, 2012). The data presented here indicate a role for Rai1 (that perhaps extends to Rat1) in regulating multiple steps of transcription genome-wide. Thus, multistage regulation of transcription by these canonical termination factors is conserved in yeast and mammals (Jimeno-González *et al*, 2010; Brannan *et al*, 2012), suggesting their importance for governing early transcription elongation, possibly through termination of improperly capped transcripts. Interestingly, our analysis of cells lacking Bye1 revealed a similar Pol II density profile as for Rai1, suggesting that they may function in a common pathway that remains to be determined.

Analysis of the 3′ end complexes revealed enrichment for many factors regulating the latter stages of elongation, 3′ end processing, and termination (Fig 6). Enrichment of Bre1 in 3′ ECs was unexpected as Bre1 is recruited to Pol II through the Paf1 complex, which we and others have observed to be enriched in 5′ ECs (Pokholok *et al*, 2002; Chen *et al*, 2015; Yu *et al*, 2015). Interestingly, we find that Bre1 regulates the dynamics of Pol II at the 3′ ends of genes, possibly by ubiquitinating histone H2B and possibly cleavage and polyadenylation factor Swd2. The importance of nucleosome modifications in transcription initiation and elongation is well established.

These findings suggest that histone modifications, in particular H2Bub, may also be important for regulating transcription termination as well.

The identification of unique 5′ and 3′ EC composition will serve as a resource for the future study of many transcription factors and can be used to uncover subgenic roles for said factors. Furthermore, the sequential IP approach with nascent RNA handles can serve as a technique to study EC composition from many genomic positions. Moving the RNA handles to a genomic loci, as opposed to the plasmid-based expression utilized here, may uncover subgenic roles for transcription factors and chromatin modifiers or remodelers not required for expression of the inducible, plasmid-based construct. By placing the stem-loop encoding sequences into different regions of genes, many different co-transcriptional processes can be analyzed. For example, MS2 sequences have been used to study splicing (Lacadie *et al*, 2006) and by inserting PP7 and MS2 sequences into intronic and exonic regions, the sequential IP approach described here could be used to isolate ECs involved in unique co-transcriptional processes. Insertion of the stem-loops into genes regulated by different promoters, enhancers, or terminators will allow exploration of how these different genetic elements regulate the composition of the elongation complex. Lastly, this approach could study genes regulated by different mechanisms, such as stress-induced genes versus housekeeping genes, which would identify factors regulating transcription elongation from different gene classes.

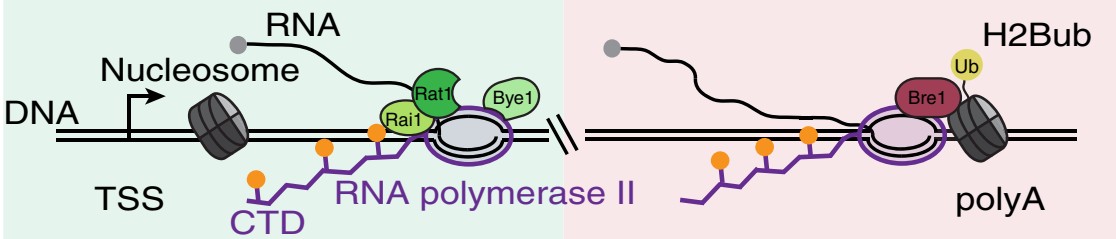

| Genic location | 5' region | 3' region |
|---|---|---|
| Transcription stage | early elongation/productive elongation | productive elongation/termination |
| CTD phosphorylation | Ser5, Ser7 phosphorylation | Ser2, Thr4 phosphorylation |
| Example Enriched Factors | Rat1/Rai1, Bye1, Spt4/Spt5, Spt6 Paf1 complex, mRNA capping complex | Bre1, Ess1, Pab1,Cft2 CCR4-Not complex 26S proteome |
| Total Enriched Factors | 159 proteins | 81 proteins |

**Figure 6.  Summary diagram of factors identified to regulate subgenic stages of transcription.**
Diagram depicting Pol II near the 5′/early elongation transcriptional phase, shaded in green and Pol II in the 3′/late elongation and termination phase of transcription, shaded in red. Factors identified by subgenic isolation of Pol II complexes and shown to regulate specific stages of transcription are displayed. Also depicted is the H2Bub modified nucleosome, which is enriched near the polyA site. Below the diagram is a table indicating genic regions where complexes were isolated from, what transcriptional stage the subgenic Pol II complexes are isolated from, the likely state of CTD phosphorylation, a sampling of factors enriched at subgenic regions during transcription, and the total number of factors enriched in each IP using the stringent cutoff.

# Materials and Methods

### Plasmid construction

The pKH202 plasmid was made by first sewing together IDT gene blocks containing the 2× PP7 stem-loop sequence from Hocine *et al* (2013), the yeast *TDH3* gene, and 2× MS2 stem-loop sequence from Hocine *et al* (2013) by overlap extension PCR. This construct (2×PP7-TDH3-2×MS2) was then cloned into p423 (Mumberg *et al*, 1994) using In-Fusion based cloning (Clontech). This placed the construct downstream of a GAL1 promoter on the 2μ plasmid. For a list of plasmids used in these study see Table EV3.

### Yeast mutant generation

Deletion mutants YSC185, YSC089, YSC072, and YSC085 were made by transforming YSC001 with PCR products of the HIS3 gene flanked by 40 bp of homology upstream and downstream of the start and stop codons for the gene of interest. Standard lithium acetate transformations were used. For a list of yeast strains used in this study see Table EV3.

### Sequential PP7-GFP and MS2-RFP IPs for mass spectrometry and NET-seq

Overnight cultures from single yeast colonies of YKH402 for GFP IPs or YKH403 for RFP IPs grown in SC-His-Ura media containing 2% glucose as the carbon source were diluted to $OD_{600} = 0.05$ into 1 l of SC-His-Ura media containing 2% raffinose as the carbon source and grown to an $OD_{600} = 0.8$–1.0. Forty percent galactose was then added to the cultures to a final concentration of 2%, and cultures were grown for an additional 1 h. After 1 h, cultures were filtered over 0.45-mm-pore-size nitrocellulose filters (Whatman). Yeast were scraped off the filter into a 10-ml syringe and forced through the syringe into liquid nitrogen, creating "noodles," as described previously (Oeffinger *et al*, 2007). Noodles were then pulverized by a ball-bearing mixer mill (Retsch MM301) for 3 min at 15 Hz for eight cycles. Sample chambers were pre-chilled in liquid nitrogen and chilled after every cycle to generate yeast "grindate". One gram of yeast grindate was resuspended in 5 ml of 1× lysis buffer [20 mM HEPES pH 7.4, 110 mM KOAc, 0.5 Triton X-100, 0.1% Tween-20, 10 mM $MnCl_2$, 8 U/ml RNasin (Promega), 1× Complete EDTA free protease inhibitor cocktail (Roche), 1× PhosStop phosphatase inhibitor (Roche)]. 660 U of RQ1 DNase (Promega) was added and the lysate incubated for 20 min on ice. Lysate was centrifuged at 20,000 *g* for 10 min at 4°C. Supernatant was added to 500 μl of ANTI-FLAG M2 affinity gel (Sigma-Aldrich) pre-equilibrated with lysis buffer and rotated at 4°C for 1 h. Immunoprecipitations were washed 3 × for 5 min at 4°C with 10 ml of 1× wash buffer (20 mM HEPES pH 7.4, 110 mM KOAc, 0.5 Triton X-100, 0.1% Tween-20, 1 mM EDTA). After washing, the beads were resuspended in 1.2 ml of wash buffer and 3.33 μg of RNase A was added to the samples and they were incubated on a rotator for 5 min at 4°C. After 5 min, the RNase A was quenched by the addition of 33 μl of RNasein (Promega) and 133 μl of 100 mM EDTA. Quenching was carried out for 60 s while rotating at 4°C. Samples were then washed with 1× wash buffer containing RNasein (8 U/ml) for 5 min once followed by two more washes for 2 min.

Complexes were eluted twice in 300 μl of 2 mg/ml 3× FLAG peptide (Sigma-Aldrich) resuspended in wash buffer by rotating at 4°C for 30 min. After elution, 600 μl of sample was added to 50 μl of GFP-Trap beads from Chromotek gtm-20 or 50 μl of RFP-Trap (Chromotek rtm-20) magnetic beads pre-equilibrated in wash buffer. For mock IPs, samples expressing the PCP-GFP construct were added to 50 μl of RFP-Trap magnetic beads. Samples were incubated for 30 min and washed 3 × by pipetting with 1 ml of wash buffer followed by a 5-min rotating wash. For mass spectrometry analysis, complexes were eluted twice by adding 300 μl of 0.1 M Gly pH 2 nutating at room temperature for 5 min. Elutions were pooled and neutralized by 60 μl of 1 M Tris pH 10 and TCA precipitated. For silver stains, TCA precipitated samples were run at 4–12% acrylamide gel and silver stained using the SliverQuest staining kit (Invitrogen). For mass spectrometry analysis, TCA precipitated samples were run on 12% acrylamide gels until the highest molecular weight proteins entered the separating gel. Samples were then stained with Coomassie blue stain, extracted from the gel, and submitted to the Taplin Mass Spectrometry Facility at Harvard Medical School for analysis. For NET-seq analysis, the beads were resuspended in 500 μl of TRIzol (Life Technologies) and incubated in a thermoshaker (Multitherm, Benchmark) for 10 min at 1,000 rpm at 25°C. RNA was subsequently isolated according the manufacturer's instructions and resuspended in a volume of 10 μl. NET-seq libraries were then generated as described below.

### NET-seq library generation

NET-seq data for WT, *rai1Δ*, and *rtt103Δ* cells were obtained from Harlen *et al* (2016). NET-seq data for *bye1Δ*, *paf1Δ*, *rtf1Δ*, and *bre1Δ* mutants were generated as follows. Cultures for NET-seq were prepared as described in Churchman and Weissman (2012). Briefly, overnight cultures from single yeast colonies grown in YPD were diluted to $OD_{600} = 0.05$ in 1 l of YPD medium and grown at 30°C shaking at 200 rpm until reaching an $OD_{600} = 0.6$–0.8. Cultures were then filtered over 0.45-mm-pore-size nitrocellulose filters (Whatman). Yeast were scraped off the filter with a spatula pre-chilled in liquid nitrogen and plunged directly into liquid nitrogen as described in Churchman and Weissman (2012). Mixer mill pulverization was performed using the conditions described above for six cycles. NET-seq IPs, isolation of nascent RNA and library construction were carried out as previously described (Churchman & Weissman, 2012). A random hexamer sequence was added to the linker to improve ligation efficiency and allow for the removal of any library biases generated from the RT step as described in Mayer *et al* (2015). After library construction, the size distribution of the library was determined by using a 2100 Bioanalyzer (Agilent) and library concentrations were determined by Qubit 2.0 fluorometer (Invitrogen). 3′ end sequencing of all samples was carried out on an Illumina NextSeq 500 with a read length of 75.

### RT–qPCR analysis

Nascent RNA from sequential Pol II-PCP IPs was isolated as described above, with each sample having a different RNase A digestion time. Digestion times at 0, 2, 5, 7, and 10 min of treatment on samples purified by the Rpb3 IP were tested. An input sample that represents whole-cell lysate was also analyzed. RT was

performed using Superscript III RT (Invitrogen) and random hexamers according to the manufacturer's instructions. After that RT samples were diluted 1:2 and 2 µl of cDNA was used for qPCR. qPCR was done using SSoFast Supermix from Bio-Rad according the manufacturer's instructions. qPCR was carried out on a Bio-Rad CFX384 Real-Time System with a C1000 Thermocycler, and Ct values were calculated using regression. qPCR primers for the 5′ region, corresponding to the PP7 stem-loop, were forward primer PP7: GCCTAGAAAGGAGCAGACGA, reverse primer PP7: CTT GAATGAACCCGGGAATAG. qPCR primers for the 3′ regions, corresponding to the MS2 stem-loops, were forward primer MS2: CGGTA CTTATTGCCAAGAAAGC, reverse primer MS2: GATGAACCCTGG AATACTGGA.

## Methods for protein sequence analysis by LC-MS/MS

### Taplin biological mass spectrometry facility

Excised gel bands were cut into approximately 1-mm$^3$ pieces. Gel pieces were then subjected to a modified in-gel trypsin digestion procedure (Shevchenko *et al*, 1996). Gel pieces were washed and dehydrated with acetonitrile for 10 min followed by removal of acetonitrile. Pieces were then completely dried in a speed-vac. Rehydration of the gel pieces was done with 50 mM ammonium bicarbonate solution containing 12.5 ng/µl modified sequencing-grade trypsin (Promega, Madison, WI, USA) at 4°C. After 45 min, the excess trypsin solution was removed and replaced with 50 mM ammonium bicarbonate solution to just cover the gel pieces. Samples were then placed in a 37°C room overnight. Peptides were later extracted by removing the ammonium bicarbonate solution, followed by one wash with a solution containing 50% acetonitrile and 1% formic acid. The extracts were then dried in a speed-vac (~1 h). The samples were then stored at 4°C until analysis. On the day of analysis, the samples were reconstituted in 5–10 µl of HPLC solvent A (2.5% acetonitrile, 0.1% formic acid). A nano-scale reverse-phase HPLC capillary column was created by packing 2.6-µm C18 spherical silica beads into a fused silica capillary (100 µm inner diameter × ~30 cm length) with a flame-drawn tip (Peng & Gygi, 2001). After equilibrating the column, each sample was loaded via a Famos auto sampler (LC Packings, San Francisco CA) onto the column. A gradient was formed, and peptides were eluted with increasing concentrations of solvent B (97.5% acetonitrile, 0.1% formic acid). As peptides eluted, they were subjected to electrospray ionization and then entered into an LTQ Orbitrap Velos Pro ion-trap mass spectrometer (Thermo Fisher Scientific, Waltham, MA, USA). Peptides were detected, isolated, and fragmented to produce a tandem mass spectrum of specific fragment ions for each peptide. Peptide sequences (and hence protein identity) were determined by matching protein databases with the acquired fragmentation pattern by the software program, Sequest (Thermo Fisher Scientific, Waltham, MA, USA) (Eng *et al*, 1994). All databases include a reversed version of all the sequences, and the data were filtered to between a one and two percent peptide FDR.

## Mass spectrometry data analysis

All mass spectrometry data analysis was done using the Perseus software (Hubner *et al*, 2010; Hubner & Mann, 2011) (http://www.coxdocs.org/doku.php?id=perseus:start) as follows. For each

run, MS1 intensities for each peptide from a given protein were summed to obtain MS1 intensity levels for each protein. The dataset for the specific triplicate IP (GFP or RFP) was loaded into Perseus along with the mock control dataset (PCP-GFP using RFP beads). Datasets were log$_2$ transformed, and each sample was normalized by subtracting the mean MS1 intensity for that dataset from each protein in the dataset. Mean normalization centers the dataset around zero without affecting the variance within each dataset. As seen in Fig EV1, mean normalization allows direct comparison between datasets by ensuring each dataset has a similar mean. As samples already have similar variances, the two datasets can be statistically analyzed by *t*-tests. Samples were then filtered for proteins present in all three specific or mock IPs. To allow for statistical analysis between the specific and mock IPs, missing values (proteins present in the specific IP but not the mock or vice versa) were imputed by assigning a value from a down-shifted normal distribution of the actual MS1 intensity distribution, representing low abundance values near noise levels, with a down-shift of 1.8 and a width of 3, as is standard (Hubner *et al*, 2010). Volcano plot analysis identified significantly enriched interactors as follows. Specifically enriched interactors were determined by comparing normalized MS1 intensities of three biological replicates of a specific IP to biological triplicates of a mock control. Proteins were only considered enriched interactors if they were present in all three specific IPs and significantly enriched as determined in the following manner. The observed fold enrichment of proteins in the specific IPs compared to mock IPs (plotted in log$_2$ scale) was compared to the negative log$_{10}$ of the *P*-value resulting from a *t*-test between each protein in the specific and mock IPs. This analysis results in a volcano plot (Hubner *et al*, 2010; Hubner & Mann, 2011). Enriched interactors are located in the upper right of the plot, with a large fold enrichment and low *P*-value. To define truly enriched proteins, a significance line corresponding to a FDR of 0.05 for high-confidence interactors or 0.1 for less stringent interactors is calculated using a permutation-based method in the mass spectrometry analysis software, Perseus (Hubner *et al*, 2010; Hubner & Mann, 2011). The SO values, which represent curve bend, were selected to minimize the number of mock-specific interactors as previously described (Hubner *et al*, 2010; Hubner & Mann, 2011). Pearson correlation values, principal component analysis, and complete linkage hierarchical clustering for mass spectrometry data were calculated using the Perseus software.

## NET-seq data analysis

NET-seq reads were aligned as follows. The adapter sequence (ATCTCGTATGCCGTCTTCTGCTTG) was removed from all reads using cutadapt with the following parameters: -O 3 -m 1 –length-tag "length=". Raw fastq files were filtered using PrinSeq (http://prinseq.sourceforge.net/) with the following parameters: -no_qual_header -min_len 7 -min_qual_mean 20 -trim_right 1 -trim_ns_right 1 -trim_qual_right 20 -trim_qual_type mean -trim_qual_window 5 -trim_qual_step 1. Random hexamer linker sequences (the first six nucleotides at the 5′ end of the read) were removed using custom python scripts but remained associated with the read, and reads were then aligned to the SacCer3 genome obtained from the *Saccharomyces* Genome Database using the TopHat2 aligner (Kim *et al*, 2013) with the following parameters: –read-mismatches

3 –read-gap-length 2 –read-edit-dist 3 –min-anchor-length 8 –splice-mismatches 1 –min-intron-length 50 –max-intron-length 1,200 –max-insertion-length 3 –max-deletion-length 3 –num-threads 4 –max-multihits 100 –library-type fr-firststrand –segment-mismatches 3 –no-coverage-search –segment-length 20 –min-coverage-intron 50 –max-coverage-intron 100,000 –min-segment-intron 50 –max-segment-intron 500,000 –b2-sensitive. To avoid any bias toward favoring annotated regions, the alignment was performed without providing a transcriptome. Reverse transcription mispriming events are identified and removed where molecular barcode sequences correspond exactly to the genomic sequence adjacent to the aligned read. For NET-seq only, the position corresponding to the 5′ end of the sequencing read (after removal of the barcode), which corresponds to the 3′ end of the nascent RNA fragment, is recorded with a custom python script using HTSeq package (Anders *et al*, 2014).

### Average profile plots and occupancy analysis

NET-seq from wild-type or mutant strains were scored around the transcription start site (TSS) and polyA sites non-overlapping genes in 1-bp bins using the deepTools program (Ramírez *et al*, 2014). Annotation for TSS and pA sites were derived from Pelechano *et al* (2013) by taking the major transcript isoform for each gene. The TSS and pA average profiles were calculated using non-overlapping protein-coding genes with an RPKM > 10 in the WT NET-seq data and that are at least 500 bp long ($n = 2,738$). For the average plots in Fig EV3, genes that did not overlap by at least 350 bp were used ($n = 1,120$). First, each NET-seq library is normalized by the number of million uniquely mapped reads. NET-seq data for each gene used in the average profile are then normalized by summing the total number of reads for that gene and dividing by the length of the window analyzed, which includes regions around the TSS (1,100 bp) and the polyA (550 bp) for a total region of 1,650 bp. After each gene is normalized, the average profile and 95% confidence interval are calculated using a sliding window of 25 base pairs, which results in average Pol II density for NET-seq around the TSS and polyA sites. To calculate the 5′ to 3′ ratio, a 5′ to 3′ score is calculated for each gene as follows. The $\log_2$ ratio of the sum of normalized reads from the 1 to 250 bp from the TSS is divided by the sum of reads from 250 bp upstream of the polyA site to the polyA site. Box plots of the distribution of 5′ to 3′ scores for each gene in WT and mutant samples are then plotted. To calculate the polyA pausing ratio, a polyA pausing score is calculated for each gene as follows. The $\log_2$ ratio of the sum of NET-seq reads in a window from 50 base pairs upstream of the polyA site to 100 base pairs downstream of the polyA site is compared to the sum of reads in a window from 200 to 50 base pairs upstream of the polyA site. Box plots of the polyA pausing ratio distributions for each sample are then plotted. Significant differences in mutant and WT samples for both the 5′ to 3′ ratio scores and polyA pausing ratio scores are then determined using two-sided *t*-tests.

### Re-analysis of histone and nucleosome data

MNase-seq data and histone ChIP-exo data were obtained from van Bakel *et al* and Rhee *et al* respectively (van Bakel *et al*, 2013; Rhee *et al*, 2014). Reads were then aligned to the SacCer3 genome obtained from the *Saccharomyces* Genome Database, using Bowtie version 1.1.1 (Langmead *et al*, 2009) with the following parameters -v 2 -M 1 –best –S. Coverage files were then generated using BEDtools version 2.23.0 with the following parameters genomeCoverageBed -bga -5 –strand. ChIP-exo data were then shifted according to the parameters given in Rhee *et al* (2014) and the strand information removed. Data were normalized by million mapped reads, and average profiles were calculated as described in the Average Profile Plots and Occupancy analysis section.

### MPA assay

Mycophenolic acid (MPA) assays were conducted as follows. Overnight cultures grown in YPD were back diluted and grown to $OD_{600}$. Serial tenfold dilutions beginning at 1 OD were then stamped on plates of YPD containing 0, 45, or 100 μg/ml of MPA. Plates were grown at 30°C and imaged every 24 h to monitor growth.

### Data availability

NET-seq data are available in the GEO repository under GEO Accession: GSE83546. Proteomics data are available in Table EV1 and in PeptideAtlas under the identifier: PASS00957.

**Expanded View** for this article is available online.

## Acknowledgements

We thank K. Trotta and E. Smith for generation of NET-seq libraries and S. Doris for spot tests. We thank the Churchman laboratory for advice and discussions and R. Singer for supplying the PP7-GFP and MCP-mCherry expression plasmids. We thank the Taplin Mass Spectrometry Facility, the Bauer Core Facility at Harvard University, and the Biopolymers Facility at Harvard Medical School for mass spectrometry assistance and next-generation sequencing, respectively. This work was supported by U.S. National Institutes of Health NHGRI grant R01HG007173 and NIGMS grant R01GM11733 to L.S.C., a Damon Runyon Dale F. Frey Award for Breakthrough Scientists (to L.S.C.), and a Burroughs Wellcome Fund Career Award at the Scientific Interface (to L.S.C.). K.M.H. was supported by U.S. National Science Foundation Graduate Research Fellowship DGE1144152.

## Author contributions

KMH and LSC designed the study. KMH generated constructs, performed immunoprecipitations, NET-seq and RT–qPCR. KMH analyzed mass spectrometry and NET-seq data. KMH and LSC wrote the manuscript.

## Conflict of interest

The authors declare that they have no conflict of interest.

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
