## [Review Process File · Molecular Systems Biology]

Subgenic Pol II interactomes identify region-specific transcription elongation regulators

Kevin M. Harlen and L. Stirling Churchman

Corresponding author: L. Stirling Churchman, Harvard Medical School

Review timeline:

Submission date:	23 August 2016
Editorial Decision:	16 September 2016
Revision received:	13 November 2016
Editorial Decision:	18 November 2016
Revision received:	29 November 2016
Accepted:	30 November 2016

Editor: Maria Polychronidou

Transaction Report:

1st Editorial Decision

16 September 2016

Thank you again for submitting your work to Molecular Systems Biology. We have now heard back from the three referees who agreed to evaluate your study. As you will see below, the reviewers think that the presented approach and findings are a valuable contribution to the field. However, they raise a number of concerns, which we would ask you to address in a revision of the study.

The recommendations of the referees are rather clear so there is no need to repeat the points/comments listed below. Please feel free to contact me in case you would like to discuss further any of the referees' comments.

REFeree REPORTS

Reviewer #1:

Harlen & Churchman investigate the interactomes of early and late Pol II transcription elongation complexes on a single gene in *S. cerevisiae*. Authors developed a strategy based on the interaction of two different viral coat proteins with their corresponding RNA stem loop recognition sequences PP7 and MS2, introduced into the 5' and 3' UTR of a plasmid-encoded and overexpressed TDH3 gene. The coat proteins are co-expressed as RFP and GFP fusions respectively and thus allow affinity purification of native elongation complexes via the nascent RNA upon transcription of the stem loop sequence. The authors claim that a short RNase treatment step can largely confine isolation via the 5' UTR stem loop sequence to early elongating complexes. Quantitative analysis of the isolated fractions by label-free mass spectrometry allows the authors to identify proteins associated with Pol II in two different transcriptional states. They identify multiple interactors during early (e.g. capping/Paf complex, Spt4/5/6) and late (e.g. CCR4-NOT complex, Ess1)

transcriptional elongation, complementing evidence presented by other groups before. Based on their mass spectrometric analysis, they propose a role for Rai1 as negative regulator of early transcription and for Bre1 during late elongation. The authors complement these findings with NET-seq analyses of wildtype yeast as well as various loss-of-function mutants. Taken together, the authors present an interesting new approach that employs RNA handles to purify nascent RNA polymerase II complexes from different stages of transcriptional elongation. Although similar strategies were already used earlier for purification of RNPs (e.g. Youngman & Green, 2005; Lacadie et al., 2006), their study is the first that applies the strategy to the purification of subgenic nascent polymerase complexes. In some parts of the submitted manuscript the authors are a bit too hasty with their conclusions (see below). I recommend that publication is considered after revision. The authors should conclusively show that their employed mass spec data normalization strategy is applicable and should address the other points below.

Major concerns

- The authors conducted global factor normalization of mass spectrometric data for each run, based on its mean protein intensity. Using this normalization strategy the authors assume that overall mean and variance of the intensity distributions is similar for all runs (e.g. Karpievitch et al., 2012, BMC Bioinformatics). Did the authors conduct appropriate statistical measures to make sure that this assumption is true? While this might be a reasonable assumption for replicates within a single condition, it is questionable whether a comparison of the different conditions (mock vs. 5' enriched and 3' enriched) can be conducted based on mean-normalized intensities. This needs to be clarified and documented in the supplement section. For example in Figure 3C the authors make a direct comparison of the normalized protein intensities in 5' and 3' conditions, under the implicit assumption of similar intensity distributions for both conditions. However, imagine if this would not be the case and the intensity distribution of the 5' condition would be generally higher than that of the 3' condition. Then a protein, which is present in both conditions at similar levels, would be on the lower end of the 5' condition distribution, but on the upper end of the 3' distribution. After mean/median normalization, the comparison of normalized intensities would then suggest strong enrichment in the 3' condition, although proteins are generally present at similar levels. This might lead to entirely wrong conclusions. Thus, the authors have to prove that their chosen normalization strategy is applicable in this context. If their label-free quantification workflow is robust enough they should generally better use non-normalized intensities (as in Hubner et al., 2011) or think about an experimental design using other normalization strategies (e.g. spike-ins etc.) or labeling approaches (see e.g. Bantscheff et al., 2012, Anal. Chem.).
- The manuscript misses vital parts about the mass spectrometric analysis within the Material and Methods section. Although the authors did not do the mass spectrometric experiments themselves, they must at least give details about how it was done by others, such as: How were the samples processed prior to mass spectrometric analysis? Which instrument was used at which settings? Which gradient was used for pre-fragmentation? Data-dependent acquisition? Resolution? Essential details about data processing, apparently with MaxQuant, are also missing. Which settings were used for the search (MS1 and MS2 tolerances, FDR thresholds for protein identification, databases, quantification settings)? The manuscript can not be considered for publication unless this is fixed.

Minor comments

- Do the authors use a single yeast strain transformed with all three plasmids (pKH202, pDZ276, PMET25MCP-mCherry), or two different yeast strains with two plasmids each (pKH202 + pDZ276 and pKH202 + PMET25MCP-mCherry) for their experiments? The authors should clarify this and list the strain(s) in the corresponding Table in the Material and Methods section.
- The authors suggest that Rai1 might act as a negative regulator during early stages of transcription elongation. They could better demonstrate this by analyzing the sensitivity of the cells to the elongation inhibitor 6-azauracil (Wu et al., 2003, Genetics). If Rai1 acts as negative elongation factor, low sensitivity of the Δ Rai1 strain to 6-AU would be expected. I strongly recommend this experiment is added as it should be doable with little effort and strengthen a key biological conclusion.
- It is not clear why the Paf1 complex (with its components e.g. Paf1 and Rtf1) is not enriched over the mock IP in the isolated 3' ECs, although Paf1 complex subunits show strong ChIP signals over the entire ORF and also have been shown to be involved in 3' end processing (Crisucci & Arndt, 2011, Genet Res Int). Can this be the result of the global factor normalization? Or might it be that

elongation complexes transcribing the artificially induced, plasmid-based template are compositionally different to those performing transcription on chromatin? At least a possible explanation must be offered.

- Lack of Bre1 should not allow recruitment of Rad6 to the promoter region during initiation and thus prevent H2Bub (Wood et al., 2003, JBC; Xiao et al., 2005, MCB). The authors state that the Bre1 Δ mutation should have stronger effects on Pol II dynamics on lowly expressed genes (p. 13, l. 6f). How do the authors come to this conclusion? Can they see such expression dependent effects in their NET-seq data for known lowly and highly expressed genes?

- As chosen by the experimental design of the study, the authors analyze the composition of Pol II elongation complexes transcribing an artificially induced, plasmid-encoded template. Thus, their findings might eventually represent an incomplete picture of what happens in the chromatin context, since the presence of chromatin remodeling factors etc. is per se not required. The authors should at least mention this at some point during the discussion.

- The authors suggest that loss of H2BK123 ubiquitination on post-polyA nucleosomes in the Bre1 Δ strain might result in accumulation of Pol II at the 3' gene end. However, they do not show that Bre1 loss leads to reduction of H2Bub on this position. Thus, their conclusion is only one of several possible and Pol II accumulation could also be brought about by other factors (e.g. Swd2), which are deregulated upon Bre1 loss. The authors could test whether Pol II accumulation is Swd2-independent e.g. by using a Swd2-deficient strain. If the pausing effects, observed in the Bre1 Δ strain, are not detected, their conclusion that H2Bub is responsible for Pol II accumulation would become more likely. The author could try this, but as it may be lengthy they could also tone down the conclusions.

- The author provide evidence that Bye1 is a negative elongation factor. This may be highlighted in the abstract as Bye1 is poorly understood yet it is one of very few elongation factors that bind directly and tightly to Pol II, competing with TFIIS for the binding site (Kinkelin et al.). Do the authors have any evidence on the interplay between TFIIS and Bye1? More generally, can the authors comment on the TFIIS elongation factor?

- the difference in NET-Seq profiles for deletion strains of Paf1 and Rtf1 should be better discussed in the light of the special role of Rtf1 as a Paf1 complex subunit.

- According to the journal guidelines, the abstract needs to be shortened down to 175 words. I also think other parts of the manuscript can be significantly shortened.

Figures and legends

- Figure 1A: It would make sense to put the bar graphs at the bottom into single extra subfigure or combine it with Fig. 1F, since they display obtained results. According to the authors, the error bar shown represents the standard error of mean for all other Pol II genes. For how many different Pol II genes were reads detected? Since the error bars are virtually invisible, the authors could display the standard error together with the mean as 'mean +/- SE' above the bar.

- Figure 1C: Could the authors please comment on what exactly the difference between the input and the 0 min bar is? It would be sufficient to explain this in the Methods section.

- Figure 1E: It is not entirely clear what is shown here and the figure legend cannot resolve this. Is it a (sequential) GFP/RFP IP of a previously Rpb3 IPed fraction, as suggested by the main text? So that the input for the GFP/RFP fraction corresponds to the bound fraction of the Rpb3 IP?

- Figure 2: The authors could emphasize in figure legends and main text that all mass spectrometric analyses (correlation etc.) were performed on the protein level (although a single peptide is the primarily measured entity by mass spectrometry). How they get from peptide to protein needs to be explained in the Methods parts (see above).

- Figure 5C: Why did the authors choose exactly these sequence windows for Pol II pausing analysis? Can they support their choice with appropriate citations? Along the same lines, how do the authors interpret the strong broadening of the ratio distribution for all loss-of-function mutants in

comparison to the wildtype?

Typos and rewording

- p. 2, l. 7: "... namely the ' 5' ' and 3' regions..."
 - p. 2, l. 14: "... dynamics during the 'later' stages of..."
 - p. 7, l. 19: "... all IPs were optimized 'to' be highly efficient..."
 - p.8, l. 14f: "Triplicate IPs of both 5' and..." This sentence is repeated almost identically in l. 17f
 - p. 9, l. 7: "... normalized MS1 intensities... "
 - p. 11, l. 16: As far as I understood the authors only re-analyzed NET-seq data from WT, Δ Rai1 and Δ Rtt103 strains, which were already published in an earlier study (Harlen et al., 2016, Cell Reports). Their phrasing "As observed previously, loss of both Rai1 and Rtt103 caused defects..." is thus misleading and suggests that not only the bioinformatics analysis, but the entire experiment was repeated in the present study. Similarly, this needs to be clarified also in the figure legends.
 - p. 12, l. 1: Must be 'Figure 4E'
 - p. 12, l. 8 and p. 15, l. 10: Based on the conducted experiments, the phrase "demonstrates a role" seems too strong. The authors should better use 'suggest' or 'indicate'
 - p.12, l. 22: Must be "... the 3' region... "
 - p. 13, l. 18: Figure 5C
 - p. 14, l. 5: "...data that reveal..."
 - p. 14, l. 7f: "... we looked 'at' a..."
 - p. 14, l. 9: Must be Figure S1B. Similarly on p. 23, l. 23. There is no reference to Figure S1A in the whole text. Thus either remove Figure or add reference.
 - p. 15, l. 20 + p. 34, l. 12: Must be 5' instead of 52 and 3' instead of 32
 - p. 17: Rephrase "... stem-loop sequence from Hocine et al. 2013 ... and 2x MS2 stem-loop sequence from (Hocine et al. 2013)..."
 - p. 19, l. 2: Must be '3.33 μ g'
 - p. 19, l. 5: Must be '4{degree sign}C'
 - p. 19, l. 13: Must be '0.1 M', and '1 M'
 - p. 34, l. 18: "hierarchical" should be in lowercase
- Figure 4A+D: 'Normalized' NET-seq reads

Reviewer #2:

The manuscript by Harlen and Churchman entitled "Subgenic Pol II interactomes identify region-specific transcription elongation regulators" reports a new strategy to isolate and characterize the composition of Pol II Elongation Complexes (ECs). This is achieved by immunoprecipitating ECs from yeast using an affinity tag on Pol II followed by RNA fragmentation, a second IP on RNA elements (PP7 in 5' UTR or MS2 in 3' UTR) and quantitative mass spectrometry of the resulting proteins. The authors identify proteins associated with ECs at the 5' and 3' ends of a model gene. The experiments and analyses are carefully and rigorously performed.

Many of the identified proteins are expected (ie. previously known to be involved in elongation) but there were some surprises. For example, the Rai1 termination factor regulates Pol II dynamics at the 5' as well as the 3' ends of genes (but the same is not true for Rtt103). The ubiquitin ligase Bre1, the 26S proteasome and the CCR4-NOT complex are all enriched at the 3' ends of genes. A limitation is that the study is only performed for a single synthetic gene, driven by a GAL promoter, on a high copy plasmid. However, a similar characterization of different types of genes (discussed in the final paragraph of the manuscript) is beyond the scope of this work. Instead, the authors address this limitation by showing that some of the proteins they identify at the 5' or 3' ends of the model gene function in a similar manner across the genome using NET-seq and previously published ChIP data.

Taken together, this is a new and exciting contribution to the field that will stimulate further biochemical and in vivo experiments aimed at understanding the contributions of the identified factors.

Major points to be considered in a revised manuscript:

1. It would be helpful to the reader to have a summary diagram at the end of the paper to bring

together the key concepts: states of transcription, Pol II CTD phosphorylation states and interactors that were identified in this study.

2. Please improve the legend for Table S1, and the description of the columns within the table.
3. Can the authors comment on why 3' end cleavage factors aren't more enriched in the 3' IP samples?
4. The authors state: "NET-seq analysis of *bye1*Δ cells revealed that Pol II density at the 5' region of the RPL26B gene was severely affected (Figure 4A), but no defects were seen at the 3' end of the gene near the polyA site." Figure 4B and 4C show increased Pol II in gene bodies and decreased Pol II near the promoter in *bye1*Δ cells. In Figure 4A, there appears to be distinct accumulation of Pol II in the 3' half of the gene, but before the polyA site *bye1*Δ cells. Are similar patterns seen on other genes?

Minor points:

- The prime symbol is corrupted in many instances
- Figure references in the text are not correct in many cases.
- Figure 1B, I can't see which band in the "M" lane correspond to the 250 kDa marker. Please mark all bands more clearly
- There are many small errors and inconsistencies in the Methods section that should be corrected (4C instead of 4{degree sign}C, ug instead of μg, etc).

Reviewer #3:

This MS reports a simple, clever approach to the isolation of transcription complexes at different locations along a highly expressed gene construct. The chief component of the MS is the description and validation of the method. The novel biological insights provided are limited but interesting. The technique clearly has the potential to be applied to many other genes, conditions and systems, and this report will be of wide interest.

I am not entirely sure that Mol Sys Biol is the ideal journal for this report, but it is good work and I would support publication with relatively minor changes.

Specific points:

- 1) Figure 3; The proteins labelled in the figure are clearly only a subset of the 5' and 3' enriched factors, and are not the most strongly enriched. What are the others? These data can presumably be extracted by analysis of data set S1 but a simple table listing the most enriched genes would help the reader understand the specificity of the purification.
- 2) Pausing at 5' end is apparently abolished, at least on the individual genes shown, by loss of either *Bye1* or *Rai1*. This seems unexpected. Are there known interactions that would suggest a common pathway? Some comment would be helpful.
- 3) Some of the changes reported are modest - e.g. the effects of *bre1*Δ at 3' ends. The statistical significance is high because many genes are involved. An indication of what fraction of genes are affected might be useful. Do the changes correlate with other features?

Reviewer #1:

Harlen & Churchman investigate the interactomes of early and late Pol II transcription elongation complexes on a single gene in *S. cerevisiae*. Authors developed a strategy based on the interaction of two different viral coat proteins with their corresponding RNA stem loop recognition sequences PP7 and MS2, introduced into the 5' and 3' UTR of a plasmid-encoded and overexpressed TDH3 gene. The coat proteins are co-expressed as RFP and GFP fusions respectively and thus allow affinity purification of native elongation complexes via the nascent RNA upon transcription of the stem loop sequence. The authors claim that a short RNase treatment step can largely confine isolation via the 5' UTR stem loop sequence to early elongating complexes. Quantitative analysis of the isolated fractions by label-free mass spectrometry allows the authors to identify proteins associated with Pol II in two different transcriptional states. They identify multiple interactors during early (e.g. capping/Paf complex, Spt4/5/6) and late (e.g. CCR4-NOT complex, Ess1) transcriptional elongation, complementing evidence presented by other groups before. Based on their mass spectrometric analysis, they propose a role for Rai1 as negative regulator of early transcription and for Bre1 during late elongation. The authors complement these findings with NET-seq analyses of wildtype yeast as well as various loss-of-function mutants. Taken together, the authors present an interesting new approach that employs RNA handles to purify nascent RNA polymerase II complexes from different stages of transcriptional elongation. Although similar strategies were already used earlier for purification of RNPs (e.g. Youngman & Green, 2005; Lacadie et al., 2006), their study is the first that applies the strategy to the purification of subgenic nascent polymerase complexes. In some parts of the submitted manuscript the authors are a bit too hasty with their conclusions (see below). I recommend that publication is considered after revision. The authors should conclusively show that their employed mass spec data normalization strategy is applicable and should address the other points below.

We thank the reviewer for the clear summary of our work and placing our method in the context of other strategies to analyze RNPs.

Major concerns

- The authors conducted global factor normalization of mass spectrometric data for each run, based on its mean protein intensity. Using this normalization strategy the authors assume that overall mean and variance of the intensity distributions is similar for all runs (e.g. Karpievitch et al., 2012, BMC Bioinformatics). Did the authors conduct appropriate statistical measures to make sure that this assumption is true?

While this might be a reasonable assumption for replicates within a single condition, it is questionable whether a comparison of the different conditions (mock vs. 5' enriched and 3' enriched) can be conducted based on mean-normalized intensities. This needs to be clarified and documented in the supplement section. For example in Figure 3C the authors make a direct comparison of the normalized protein intensities in 5' and 3' conditions, under the implicit assumption of similar intensity distributions for both conditions. However, imagine if this would not be the case and the intensity distribution of the 5' condition would be generally higher than that of the 3' condition. Then a protein, which is present in both conditions at similar levels, would be on the lower end of the 5' condition distribution, but on the upper end of the 3' distribution. After mean/median normalization, the comparison of normalized intensities would then suggest strong enrichment in the 3' condition, although proteins are generally present at similar levels. This might lead to entirely wrong conclusions. Thus, the authors have to prove that their chosen normalization strategy is applicable in this context. If their label-free quantification workflow is robust enough they should generally better use non-normalized intensities (as in Hubner et al., 2011) or think about an experimental design using other normalization strategies (e.g. spike-ins etc.) or labeling approaches (see e.g. Bantscheff et al., 2012, Anal. Chem.).

This is an important point. Normalization by mean protein intensity assumes that the bulk of analyzed proteins are at similar relative abundance across samples. Here we are analyzing the results of a sequential IP, where the first IP isolates all Pol II. The second IP enriches for Pol II subpopulations that each contain the 12 core Pol II subunits and the many proteins that remain associated with Pol II across gene bodies. Thus, these subpopulations are expected to

be fairly similar and mass spectrometric analysis of the second IP is expected to have comparable mean protein intensities. Indeed, we see that all samples have similar variances and the 5' and 3' (GFP and RFP) IPs have similar means. Mass spectrometric analysis of the mock IP identified fewer proteins than the 5' and 3' IPs, which explains why the mock IPs results in higher MS1 intensities. We thank the reviewer for the suggestion to add this analysis to the manuscript, which is now shown in Extended View figure 1 (Figure EV1). We also added further information about mean normalization to the Materials and Methods section.

- The manuscript misses vital parts about the mass spectrometric analysis within the Material and Methods section. Although the authors did not do the mass spectrometric experiments themselves, they must at least give details about how it was done by others, such as: How were the samples processed prior to mass spectrometric analysis? Which instrument was used at which settings? Which gradient was used for pre-fragmentation? Data-dependent acquisition? Resolution? Essential details about data processing, apparently with MaxQuant, are also missing. Which settings were used for the search (MS1 and MS2 tolerances, FDR thresholds for protein identification, databases, quantification settings)? The manuscript can not be considered for publication unless this is fixed.

We have now extended the Materials and Methods section to include the answers to these questions. We thank the reviewer for pointing out the omission.

Minor comments

-Do the authors use a single yeast strain transformed with all three plasmids (pKH202, pDZ276, PMET25MCP-mCherry), or two different yeast strains with two plasmids each (pKH202 + pDZ276 and pKH202 + PMET25MCP-mCherry) for their experiments? The authors should clarify this and list the strain(s) in the corresponding Table in the Material and Methods section.

We have clarified the plasmids used in these studies in the Materials and Methods section and strain tables.

- The authors suggest that Rai1 might act as a negative regulator during early stages of transcription elongation. They could better demonstrate this by analyzing the sensitivity of the cells to the elongation inhibitor 6-azauracil (Wu et al., 2003, Genetics). If Rai1 acts as negative elongation factor, low sensitivity of the Δ Rai1 strain to 6-AU would be expected. I strongly recommend this experiment is added as it should be doable with little effort and strengthen a key biological conclusion.

We thank the reviewer for the suggested experiment. We assayed the effects of mycophenolic acid (MPA), which is a transcription elongation inhibitor akin to 6-AU (Figure EV2). As expected, both *rai1* Δ and *bye1* Δ mutants display slightly decreased sensitivity to MPA.

- It is not clear why the Paf1 complex (with its components e.g. Paf1 and Rtf1) is not enriched over the mock IP in the isolated 3' ECs, although Paf1 complex subunits show strong ChIP signals over the entire ORF and also have been shown to be involved in 3' end processing (Crisucci & Arndt, 2011, Genet Res Int). Can this be the result of the global factor normalization? Or might it be that elongation complexes transcribing the artificially induced, plasmid-based template are compositionally different to those performing transcription on chromatin? At least a possible explanation must be offered.

The Paf1 complex members, Paf1 and Ctr9, show a modest enrichment of 1.5 fold in the 3' interactome with P values of 0.018 and 0.013. Our stringent false discovery rate of 0.05 is the reason why these factors are not considered enriched in the 3' dataset. False negatives is a cost of a stringent false discovery rate, which is why we focused on factors that show enrichment rather than factors that are not enriched. We have added this information into the text on pg 14.

- Lack of Bre1 should not allow recruitment of Rad6 to the promoter region during initiation and thus prevent H2Bub (Wood et al., 2003, JBC; Xiao et al., 2005, MCB). The authors state that the Bre1 Δ mutation should have stronger effects on Pol II dynamics on lowly expressed genes (p. 13, l. 6f). How do the authors come to this conclusion? Can they see such expression dependent effects in their NET-seq data for known lowly and highly expressed genes?

We thank the reviewer for bringing this to our attention. Our statements on this were not clear and we have not clarified our discussion of our findings in the context of Batta et al., 2011 and Xiao et al., 2005 on pg. 14. We apologize for the misleading statement.

- As chosen by the experimental design of the study, the authors analyze the composition of Pol II elongation complexes transcribing an artificially induced, plasmid-encoded template. Thus, their findings might eventually represent an incomplete picture of what happens in the chromatin context, since the presence of chromatin remodeling factors etc. is per se not required. The authors should at least mention this at some point during the discussion.

This point has been added to the text on pg. 18

- The authors suggest that loss of H2BK123 ubiquitination on post-polyA nucleosomes in the Bre1 Δ strain might result in accumulation of Pol II at the 3' gene end. However, they do not show that Bre1 loss leads to reduction of H2Bub on this position. Thus, their conclusion is only one of several possible and Pol II accumulation could also be brought about by other factors (e.g. Swd2), which are deregulated upon Bre1 loss. The authors could test whether Pol II accumulation is Swd2-independent e.g. by using a Swd2-deficient strain. If the pausing effects, observed in the Bre1 Δ strain, are not detected, their conclusion that H2Bub is responsible for Pol II accumulation would become more likely. The author could try this, but as it may be lengthy they could also tone down the conclusions.

We have altered the text on pg. 16 to suggest alternative mechanisms.

- The author provide evidence that Bye1 is a negative elongation factor. This may be highlighted in the abstract as is poorly understood yet it is one of very few elongation factors that bind directly and tightly to Pol II, competing with TFIIS for the binding site (Kinkelin et al.). Do the authors have any evidence on the interplay between TFIIS and Bye1? More generally, can the authors comment on the TFIIS elongation factor?

The role of Bye1 in transcription and its interplay with Dst1 (TFIIS) is indeed interesting and we thank the reviewer for the suggestion to add the Bye1 findings to the abstract and we have done so. We agree that the interplay between Bye1 and TFIIS will be important to determine, but we hope that the reviewer can agree that it would be outside the scope of this manuscript.

- the difference in NET-Seq profiles for deletion strains of Paf1 and Rtf1 should be better discussed in the light of the special role of Rtf1 as a Paf1 complex subunit.

We have added a more thorough discussion of Paf1 and Rtf1 on pgs. 14.

- According to the journal guidelines, the abstract needs to be shortened down to 175 words. I also think other parts of the manuscript can be significantly shortened.

We thank the reviewer for noticing and have worked to convey our findings in a more succinct manner throughout the manuscript.

Figures and legends

- Figure 1A: It would make sense to put the bar graphs at the bottom into single extra subfigure or combine it with Fig. 1F, since they display obtained results. According to the authors, the error bar shown represents the standard error of mean for all other Pol II genes. For how many different Pol II genes were reads detected? Since the error bars are virtually invisible, the authors could display the standard error together with the mean as 'mean +/- SE' above the bar.

We have created a new subfigure with the bar graphs as altered the display of the mean and standard deviation to be values listed above the bars. We have also added the number of genes detected in each IP to the figure legend. 186 Pol II genes were detected in the GFP IPs and 1,118 Pol II genes were detected in the RFP IPs.

- Figure 1C: Could the authors please comment on what exactly the difference between the input

and the 0 min bar is? It would be sufficient to explain this in the Methods section.

The input sample represent whole cell lysate while the 0min is after the Rpb3 IP with no RNaseA fragmentation. We have added this information to Materials and Methods and the figure legend.

- Figure 1E: It is not entirely clear what is shown here and the figure legend cannot resolve this. Is it a (sequential) GFP/RFP IP of a previously Rpb3 IPed fraction, as suggested by the main text? So that the input for the GFP/RFP fraction corresponds to the bound fraction of the Rpb3 IP?

We thank the reviewer for bringing this to our attention. The reviewer is correct that the input for the GFP and RFP IPs corresponds to the bound fraction of the Rpb3 IP, demonstrating that in both the Rpb3 IP and GFP/RFP IPs are highly efficient. We have clarified this in the Figure 1 legend.

- Figure 2: The authors could emphasize in figure legends and main text that all mass spectrometric analyses (correlation etc.) were performed on the protein level (although a single peptide is the primarily measured entity by mass spectrometry). How they get from peptide to protein needs to be explained in the Methods parts (see above).

We thank the author for this suggestion and have clarified this in both the Materials and Methods as well as figure legends. We mistakenly stated that "MS1 intensities for each protein in each run were summed." Instead, for each run MS1 intensities for each peptide from a given protein were summed to obtain MS1 intensity levels for each protein. We have updated the text.

- Figure 5C: Why did the authors choose exactly these sequence windows for Pol II pausing analysis? Can they support their choice with appropriate citations?

These windows were chosen based on wild-type NET-seq data that shows Pol II pausing from 50bp upstream of the polyA site until 200bp downstream of the polyA site. A similar pausing window was also observed in Schaughency et al., 2014 *Plos Genetics*, where similar Pol II pausing was observed from -50 to +200 bp around the polyA site.

Along the same lines, how do the authors interpret the strong broadening of the ratio distribution for all loss-of-function mutants in comparison to the wildtype?

The width increase across the ratio distributions of loss-of-function mutants reflects the variability of each factor's impact across the genome.

Typos and rewording

- p. 2, l. 7: "... namely the ' 5' ' and 3' regions..."
- p. 2, l. 14: "... dynamics during the 'later' stages of..."
- p. 7, l. 19: "... all IPs were optimized 'to' be highly efficient..."
- p.8, l. 14f: "Triplicate IPs of both 5' and..." This sentence is repeated almost identically in l. 17f
- p. 9, l. 7: "... normalized MS1 intensities... "
- p. 11, l. 16: As far as I understood the authors only re-analyzed NET-seq data from WT, Δ Rai1 and Δ Rtt103 strains, which were already published in an earlier study (Harlen et al., 2016, Cell Reports). Their phrasing "As observed previously, loss of both Rai1 and Rtt103 caused defects..." is thus misleading and suggests that not only the bioinformatics analysis, but the entire experiment was repeated in the present study. Similarly, this needs to be clarified also in the figure legends.

We have clarified this in the text, see pg. 12 and the legend of Figure 4.

- p. 12, l. 1: Must be 'Figure 4E'
- p. 12, l. 8 and p. 15, l. 10: Based on the conducted experiments, the phrase "demonstrates a role" seems too strong. The authors should better use 'suggest' or 'indicate'
- p.12, l. 22: Must be "... the 3' region... "
- p. 13, l. 18: Figure 5C
- p. 14, l. 5: "...data that reveal..."

- p. 14, l. 7f: "... we looked 'at' a..."
 - p. 14, l. 9: Must be Figure S1B. Similarly on p. 23, l. 23. There is no reference to Figure S1A in the whole text. Thus either remove Figure or add reference.
 - p. 15, l. 20 + p. 34, l. 12: Must be 5' instead of 52 and 3' instead of 32
 - p. 17: Rephrase "... stem-loop sequence from Hocine et al. 2013 ... and 2x MS2 stem-loop sequence from (Hocine et al. 2013)..."
 - p. 19, l. 2: Must be '3.33 μ g'
 - p. 19, l. 5: Must be '4[°]C'
 - p. 19, l. 13: Must be '0.1 M', and '1 M'
 - p. 34, l. 18: "hierarchical" should be in lowercase
- Figure 4A+D: 'Normalized' NET-seq reads

We thank the reviewer for the thorough evaluation of the manuscript and have corrected these typographical errors.

Reviewer #2:

The manuscript by Harlen and Churchman entitled "Subgenomic Pol II interactomes identify region-specific transcription elongation regulators" reports a new strategy to isolate and characterize the composition of Pol II Elongation Complexes (ECs). This is achieved by immunoprecipitating ECs from yeast using an affinity tag on Pol II followed by RNA fragmentation, a second IP on RNA elements (PP7 in 5' UTR or MS2 in 3' UTR) and quantitative mass spectrometry of the resulting proteins. The authors identify proteins associated with ECs at the 5' and 3' ends of a model gene. The experiments and analyses are carefully and rigorously performed.

Many of the identified proteins are expected (ie. previously known to be involved in elongation) but there were some surprises. For example, the Rai1 termination factor regulates Pol II dynamics at the 5' as well as the 3' ends of genes (but the same is not true for Rtt103). The ubiquitin ligase Bre1, the 26S proteasome and the CCR4-NOT complex are all enriched at the 3' ends of genes. A limitation is that the study is only performed for a single synthetic gene, driven by a GAL promoter, on a high copy plasmid. However, a similar characterization of different types of genes (discussed in the final paragraph of the manuscript) is beyond the scope of this work. Instead, the authors address this limitation by showing that some of the proteins they identify at the 5' or 3' ends of the model gene function in a similar manner across the genome using NET-seq and previously published CHIP data.

Taken together, this is a new and exciting contribution to the field that will stimulate further biochemical and in vivo experiments aimed at understanding the contributions of the identified factors.

We thank the reviewer for the positive comments and for the enthusiasm.

Major points to be considered in a revised manuscript:

1. It would be helpful to the reader to have a summary diagram at the end of the paper to bring together the key concepts: states of transcription, Pol II CTD phosphorylation states and interactors that were identified in this study.

We agree that a summary would be helpful and we have added Figure 6 summarizing the results.

2. Please improve the legend for Table S1, and the description of the columns within the table.

We have improved labeling within Table EV1 as well as the legend.

3. Can the authors comment on why 3' end cleavage factors aren't more enriched in the 3' IP samples?

A number of reasons could explain why our 3' end purification and mass spectrometry

analysis did not identify many 3' end cleavage factors. 1) The stringency of our statistical analysis will certainly produce false negatives. For example, two cleavage factors, Ctf2 and Pcf11, are enriched in the 3' IP, but not highly enough to pass our statistical filters. 2) Cleavage factors are likely transient interactors. 3) Our RNAase treatment may elute off some interactors. We have now added a discussion of the low enrichment of 3' end processing factors to the manuscript (pg 10).

4. The authors state: "NET-seq analysis of *bye1*Δ cells revealed that Pol II density at the 5' region of the RPL26B gene was severely affected (Figure 4A), but no defects were seen at the 3' end of the gene near the polyA site." Figure 4B and 4C show increased Pol II in gene bodies and decreased Pol II near the promoter in *bye1*Δ cells. In Figure 4A, there appears to be distinct accumulation of Pol II in the 3' half of the gene, but before the polyA site *bye1*Δ cells. Are similar patterns seen on other genes?

We thank the reviewer for noticing an extended range of *Bye1* impact. Similar patterns are indeed observed on other genes. We have modified the text to describe our *bye1A* results more completely in pg. 11.

Minor points:

- The prime symbol is corrupted in many instances
- Figure references in the text are not correct in many cases.
- Figure 1B, I can't see which band in the "M" lane correspond to the 250 kDa marker. Please mark all bands more clearly
- There are many small errors and inconsistencies in the Methods section that should be corrected (4C instead of 4{degree sign}C, ug instead of μg, etc).

We thank the reviewer for bringing these errors to our attention. We have now corrected these errors throughout the text.

Reviewer #3:

This MS reports a simple, clever approach to the isolation of transcription complexes at different locations along a highly expressed gene construct. The chief component of the MS is the description and validation of the method. The novel biological insights provided are limited but interesting. The technique clearly has the potential to be applied to many other genes, conditions and systems, and this report will be of wide interest.

I am not entirely sure that Mol Sys Biol is the ideal journal for this report, but it is good work and I would support publication with relatively minor changes.

We thank the reviewer for the positive comments and appreciation for the potential of our approach.

Specific points:

1) Figure 3; The proteins labelled in the figure are clearly only a subset of the 5' and 3' enriched factors, and are not the most strongly enriched. What are the others? These data can presumably be extracted by analysis of data set S1 but a simple table listing the most enriched genes would help the reader understand the specificity of the purification.

We agree and have added a supplemental table, Table EV2 listing identified factors by their enrichment.

2) Pausing at 5' end is apparently abolished, at least on the individual genes shown, by loss of either *Bye1* or *Rai1*. This seems unexpected. Are there known interactions that would suggest a common pathway? Some comment would be helpful.

To the best of our knowledge, there are no known connections between *Bye1* and *Rai1*. We mention this on pg. 17

3) Some of the changes reported are modest - e.g. the effects of *bre1Δ* at 3' ends. The statistical significance is high because many genes are involved. An indication of what fraction of genes are affected might be useful. Do the changes correlate with other features?

We have added an additional analysis to Figure 5, which demonstrates that 60% of genes show increased polyA pausing in the *bre1Δ* mutant when compared to wild-type. We did not identify any distinct features separating the affected and unaffected genes.

2nd Editorial Decision

18 November 2016

Thank you again for submitting your work to Molecular Systems Biology. We have now heard back from reviewer #1 who was asked to evaluate your revised study. As you will see below, this reviewer is satisfied with the modifications made and thinks that the study is now suitable for publication in Molecular Systems biology.

Before we formally accept your paper, we would ask you to address some minor editorial issues listed below.

REFEREE REPORTS

Reviewer #1:

The authors have addressed my concerns and the manuscript is suited for publication in MSB.

Corresponding Author Name: Stirling Churchman
Journal Submitted to: Molecular Systems Biology
Manuscript Number: MSB-16-7279